# The SPOC proteins DIDO3 and PHF3 co-regulate gene expression and neuronal differentiation

Johannes Benedum [1,2,3,4], Vedran Franke [5], Lisa-Marie Appel [1,2,3], Lena Walch[1], Melania Bruno [1], Rebecca Schneeweiss[1], Juliane Gruber[1], Helena Oberndorfer[1], Emma Frank[1], Xué Strobl[1,4], Anton Polyansky [6], Bojan Zagrovic [6], Altuna Akalin [5] & Dea Slade [1,2,3] ✉

Transcription is regulated by a multitude of activators and repressors, which bind to the RNA polymerase II (Pol II) machinery and modulate its progression. Death-inducer obliterator 3 (DIDO3) and PHD finger protein 3 (PHF3) are paralogue proteins that regulate transcription elongation by docking onto phosphorylated serine-2 in the C-terminal domain (CTD) of Pol II through their SPOC domains. Here, we show that DIDO3 and PHF3 form a complex that bridges the Pol II elongation machinery with chromatin and RNA processing factors and tethers Pol II in a phase-separated microenvironment. Their SPOC domains and C-terminal intrinsically disordered regions are critical for transcription regulation. PHF3 and DIDO exert cooperative and antagonistic effects on the expression of neuronal genes and are both essential for neuronal differentiation. In the absence of PHF3, DIDO3 is upregulated as a compensatory mechanism. In addition to shared gene targets, DIDO specifically regulates genes required for lipid metabolism. Collectively, our work reveals multiple layers of gene expression regulation by the DIDO3 and PHF3 paralogues, which have specific, co-regulatory and redundant functions in transcription.

Mammalian gene expression networks comprise a multitude of effectors that often act cooperatively and redundantly to fine tune specific transcriptional programmes[1]. These effectors modulate chromatin structure, directly bind and regulate Pol II activity, or modify and process RNA. Pol II CTD is an important platform for the recruitment of different effectors that recognize specific CTD phosphorylation marks[2].

PHD finger protein 3 (PHF3) was recently identified as a mammalian regulator of transcription elongation and mRNA stability, which specifically recognizes phospho-Ser2 Pol II CTD through its Spen Paralogue and Orthologue C-terminal (SPOC) domain[3]. PHF3 knockout or SPOC deletion leads to a prominent derepression of neuronal genes in differentiated cells. In mouse embryonic stem cells (mESCs), loss of Phf3 leads to precocious expression of neuronal genes and impairs neuronal differentiation. The mammalian *DIDO1* gene expresses three isoforms: DIDO1, DIDO2 and DIDO3. The longest Isoform, DIDO3, shares the same domain architecture with PHF3 that consists of a Plant homeodomain (PHD), TFIIS-like domain (TLD) and

[1]Department of Medical Biochemistry, Medical University of Vienna, Max Perutz Labs, Vienna Biocenter, Vienna, Austria. [2]Department of Radiation Oncology, Medical University of Vienna, Vienna, Austria. [3]Comprehensive Cancer Center, Medical University of Vienna, Vienna, Austria. [4]Vienna Biocenter PhD Program, a Doctoral School of the University of Vienna and Medical University of Vienna, Vienna, Austria. [5]The Berlin Institute for Medical Systems Biology, Max Delbrück Center, Berlin, Germany. [6]Department of Structural and Computational Biology, Max Perutz Labs, University of Vienna, Vienna Biocenter, Vienna, Austria. ✉e-mail: dea.slade@maxperutzlabs.ac.at

the SPOC domain (Fig. 1a). The two shorter DIDO isoforms, DIDO1 and DIDO2, lack the SPOC domain, and the smallest isoform, DIDO1, also lacks TLD and only contains PHD (Fig. 1a). While PHF3 PHD cannot bind histones, DIDO PHD specifically recognizes H3K4me3 as a mark of active transcription[4,5]. PHF3 and DIDO TLD share homology with the

transcription factor TFIIS but lack domain III that stimulates RNA cleavage by Pol II[6]. PHF3 can outcompete TFIIS from Pol II in a TLD-dependent manner and thereby confer transcriptional repression[3]. Due to high similarity between their TLDs, DIDO3 is likely to function in a similar manner, although this has not yet been experimentally

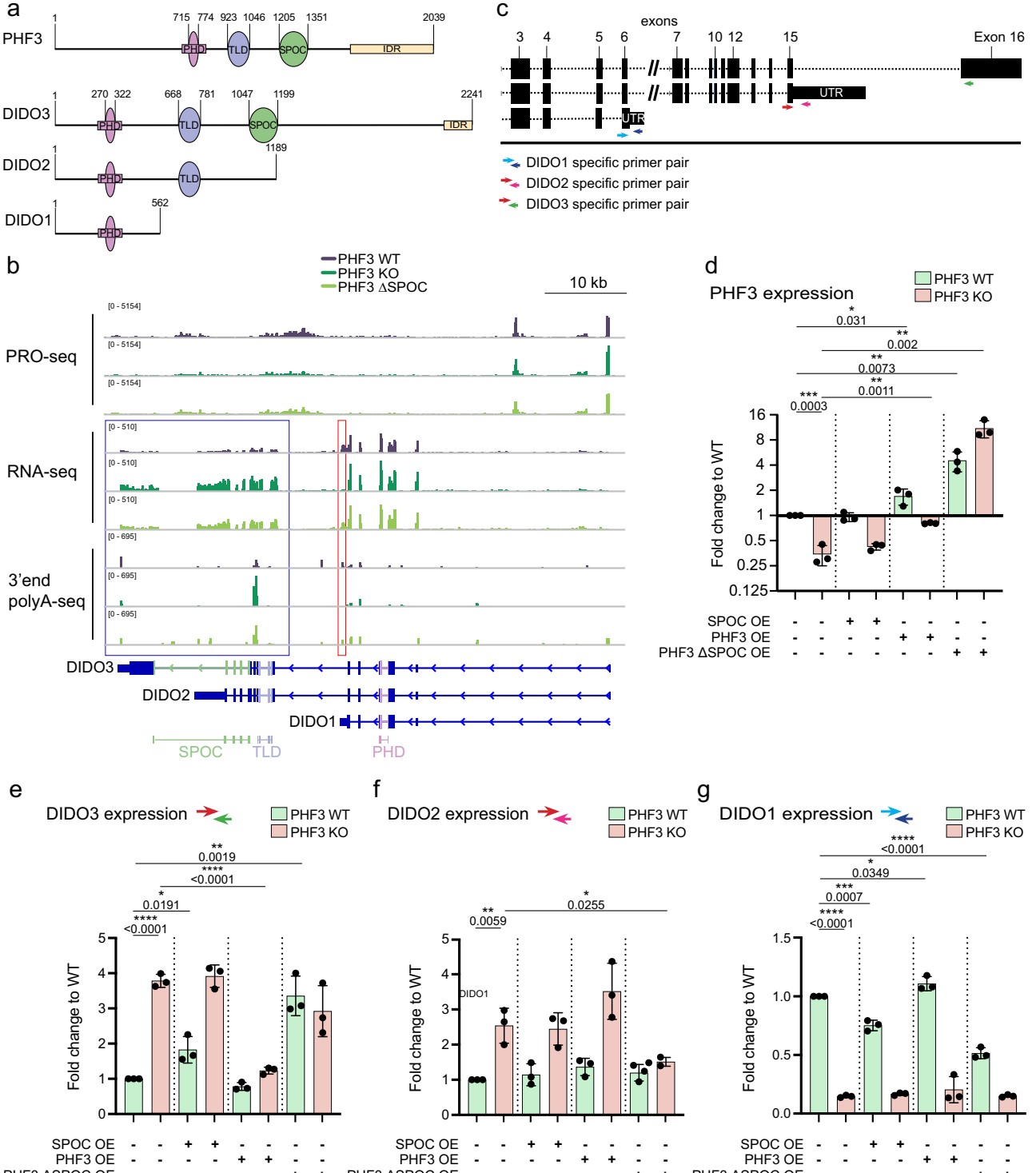

**Fig. 1 | DIDO isoform switching in PHF3 KO and ΔSPOC. a** Domain architecture of PHF3, DIDO3, DIDO2 and DIDO1. **b** Genome browser snapshots showing RNA-seq, PRO-seq and 3′end polyA-seq reads for DIDO. Long DIDO isoform comprising the SPOC domain is upregulated in PHF3 KO and ΔSPOC cells. **c** RT-qPCR strategy using primers specific for the three DIDO splice variants. **d**–**g** RT-qPCR analysis of **d** PHF3 expression, **e** DIDO3 expression, **f** DIDO2 expression and **g** DIDO1 expression. cDNA

was reverse transcribed using oligo-dT primers. qPCR primers were designed to span exons. Gene expression was normalized to GAPDH and differential expression levels were expressed as a fold change compared to PHF3 WT. Three biologically independent experiments were performed. Data are presented as mean values ± standard deviation. One-tailed, two-sample equal variance t-test was used to determine p-values. Source data are provided as a Source Data file.

explored. Like PHF3, DIDO3 SPOC preferentially recognizes phospho-Ser2 Pol II CTD but with 10-fold lower affinity[7]. In addition to these three structurally defined regions, PHF3 and DIDO3 contain large N- and C-terminal disordered regions. PHF3 was shown to sequester Ser2-phosphorylated Pol II within liquid-like condensates in vitro and colocalize within Pol II clusters in cells[3], but the region that confers such properties and whether they are shared by DIDO3 remained unknown. The very C-terminus of PHF3 and DIDO3 contains a low complexity domain enriched with the amino acids arginine, glutamic acid and aspartic acid with a near to equal distribution of opposite charges. Arginine-enriched mixed-charge domains (MCD) have been shown to form condensates and drive nuclear speckle assembly[8]. Liquid-liquid phase separation (LLPS) has been implicated in transcription regulation as a means of locally increasing the concentration of different factors at specific steps during the transcription cycle. Pol II and many transcription factors have intrinsically disordered regions (IDRs), which are prone to engage in multivalent protein/DNA/RNA interactions that drive clustering through LLPS[9–11].

PHF3 knock-out mice are viable but suffer from neuronal dysfunction (www.mousephenotype.org). Homozygous C-terminal truncation of DIDO3 (exon 16) causes early embryonic lethality in mice due to high levels of DNA damage, reduced proliferation rates, and increased apoptosis[12]. The C-terminal region of DIDO3 was shown to interact with splicing factors and recruit them to Pol II, its loss resulting in increased exon skipping, differential isoform expression, 3'UTR lengthening, and impaired differentiation[12–15]. DIDO3 also interacts with the helicase DHX9 and thereby regulates R-loop levels at transcription termination sites[13]. Mice with an N-terminal truncation of the DIDO isoforms are viable but develop myeloid neoplasm, while patients suffering from rare myelodysplastic/myeloproliferative diseases (MDS/MPDs) show abnormal DIDO isoform expression[16]. Combined heterozygous N- and C-terminal truncations of DIDO3 cause perinatal lethality with craniofacial and neurobehavioral abnormalities[17]. Conditional DIDO3 (exon 16) deletion in adult mice results in mild hepatitis, testicular degeneration, and progressive ataxia[18]. Overall, both paralogues are required for neuronal functions but DIDO3 seems to be essential during development.

In this study we examined the functional connections between DIDO3 and PHF3 paralogues in transcription regulation in human cells. We generated DIDO3 and PHF3 knock-out, ΔSPOC and ΔIDR cell lines and performed RNA-seq to determine how PHF3 and DIDO regulate gene expression, how the loss of one paralogue affects the expression of the other and how DIDO SPOC and IDR domains contribute to transcription regulation. We found that PHF3 regulates DIDO isoform expression by stabilizing the short DIDO1 isoform and downregulating the long DIDO3 isoform. PHF3 depletion induces isoform switching from the short DIDO1 to the long DIDO3, which can compensate for PHF3 loss. In addition to compensatory functions, DIDO3 and PHF3 also show agonistic and antagonistic effects in gene regulation. Neuronal genes featured as common targets of the two paralogues, the depletion of which results in failed neuronal differentiation of mESCs.

## Results
### PHF3 loss results in the upregulation of the SPOC-containing DIDO3 isoform
The PHF3 paralogue DIDO switches isoforms during embryonic stem cell (ESC) differentiation, from the long isoform DIDO3 that contains the SPOC domain and promotes the stem cell state to the short isoform DIDO1 that lacks SPOC and triggers differentiation[15] (Fig. 1a). The DIDO1 isoform was also predominantly expressed in WT HEK293T cells (Fig. 1b). Intriguingly, loss of PHF3 or its SPOC domain led to isoform switching from DIDO1 to DIDO3 (Fig. 1b). We ruled out the possibility that DIDO3 is upregulated due to transcriptional adaptation[19], as DIDO3 nascent transcripts based on PRO-seq analysis are not upregulated in PHF3 KO and ΔSPOC (Fig. 1b). 3' end sequencing of

polyadenylated mRNAs showed loss of DIDO1-specific polyadenylation site (PAS) in PHF3 KO, suggesting that PHF3 regulates DIDO isoform expression by regulating the choice of PAS (Fig. 1b). Upregulation of the long DIDO3 isoform in PHF3 KO suggests that it might partially compensate for the loss of PHF3.

To understand the underlying mechanism of DIDO3 upregulation, we generated stable cell lines expressing full-length PHF3, PHF3 ΔSPOC or the PHF3 SPOC domain alone in PHF3 KO or WT cells and examined the expression of DIDO isoforms by RT-qPCR (Supplementary Fig. 1 and Fig. 1c). Reintroduction of full-length PHF3 into PHF3 KO cells (PHF3 KO/PHF3 OE) led to restoration of PHF3 expression close to WT levels (Fig. 1d). RT-qPCR analysis confirmed DIDO3 and DIDO2 upregulation and DIDO1 downregulation in PHF3 KO cells (Fig. 1e–g), as previously revealed by RNA-seq. DIDO3 was also upregulated at the protein level in PHF3 KO (Supplementary Fig. 16b). The expression of full-length PHF3 rescued the upregulation of DIDO3 (Fig. 1e) but not the upregulation of DIDO2 (Fig. 1f) or the downregulation of DIDO1 (Fig. 1g) in PHF3 KO background. In contrast, the introduction of PHF3 ΔSPOC into PHF3 KO rescued the upregulation of DIDO2 (Fig. 1f) but not DIDO3 upregulation (Fig. 1e) or DIDO1 downregulation (Fig. 1g). Overexpression of the SPOC domain alone had no effect on DIDO expression levels (Fig. 1e–g). Surprisingly, the overexpression of PHF3 ΔSPOC in WT cells resulted in DIDO3 upregulation (Fig. 1e) and DIDO1 downregulation (Fig. 1g) similar to PHF3 KO, suggesting that PHF3 ΔSPOC may act as a dominant negative mutant.

Overall, these results suggest that PHF3 directly regulates DIDO3 expression levels and that the PHF3 SPOC domain is important in the cooperative or complementary functions of DIDO3 and PHF3 given that (i) only full length PHF3 completely rescues DIDO3 expression in PHF3 KO, and (ii) overexpression of PHF3 ΔSPOC in WT cells has a dominant negative effect on DIDO3 expression levels.

### DIDO3 interacts with chromatin-associated proteins and RNA polymerase II
To understand how PHF3 and DIDO3 function could be interconnected at the molecular level, we compared the interactome of DIDO isoforms and PHF3 by co-immunoprecipitation (co-IP) followed by mass spectrometry (Fig. 2 and Supplementary Data 1). We also included N-terminally truncated DIDO3 variants lacking exon 3 (FLAG-DIDO1 ΔN, FLAG-DIDO2 ΔN, FLAG-DIDO3 ΔN) as well as a DIDO3 SPOC deletion mutant (FLAG-DIDO3 ΔSPOC) to determine SPOC-dependent interactions of DIDO3 (Fig. 2a). All constructs showed nuclear expression according to immunofluorescence analysis (Supplementary Fig. 2a). Despite lower expression levels compared to other isoforms, FLAG-DIDO2 and FLAG-DIDO2 ΔN showed the same, albeit weaker, interactions as FLAG-DIDO1 and FLAG-DIDO1 ΔN respectively (Fig. 2d).

Histone variants (H1, H2, macroH2A.1, H2AZ, H2AX) ranked among the highest confidence interactors for all full length DIDO isoforms and DIDO ΔSPOC (Fig. 2b, d, Supplementary Figs. 2b, 3a, 4a, d and Supplementary Data 1). Interaction with histones and other chromatin associated proteins was lost for the N-terminally truncated DIDO versions, which partly lack the PHD domain (Fig. 2d, Supplementary Fig. 3c, d, Supplementary Fig. 5, Supplementary Data 1). Among N-terminal interactors we also identified chromatin associated factors (DEK, CBX, HIRIP3, BANF1, NAP1L1, HP1BP3, FACT complex) as well as several proteins implicated in DNA repair and DNA damage signaling (PARP1, PARP2, YB1, XRCC1, POLB) (Fig. 2d, Supplementary Fig. 3c, d).

Apart from histones and chromatin associated proteins, DIDO3 was found to interact strongly with Pol II and transcription elongation factors (DSIF, SPT6, PAF1C, FACT) as well as several phosphatases and kinases (CK2, CDK11B, PP2A, PP1-PNUTS) (Fig. 2b, d and Supplementary Fig. 2b, c). Interaction with Pol II was retained for DIDO3 ΔN but lost for DIDO3 ΔSPOC (Fig. 2d, Supplementary Figs. 2c, 3a, b, 5). The SPOC domain of DIDO3 is responsible for its binding to Pol II pSer2

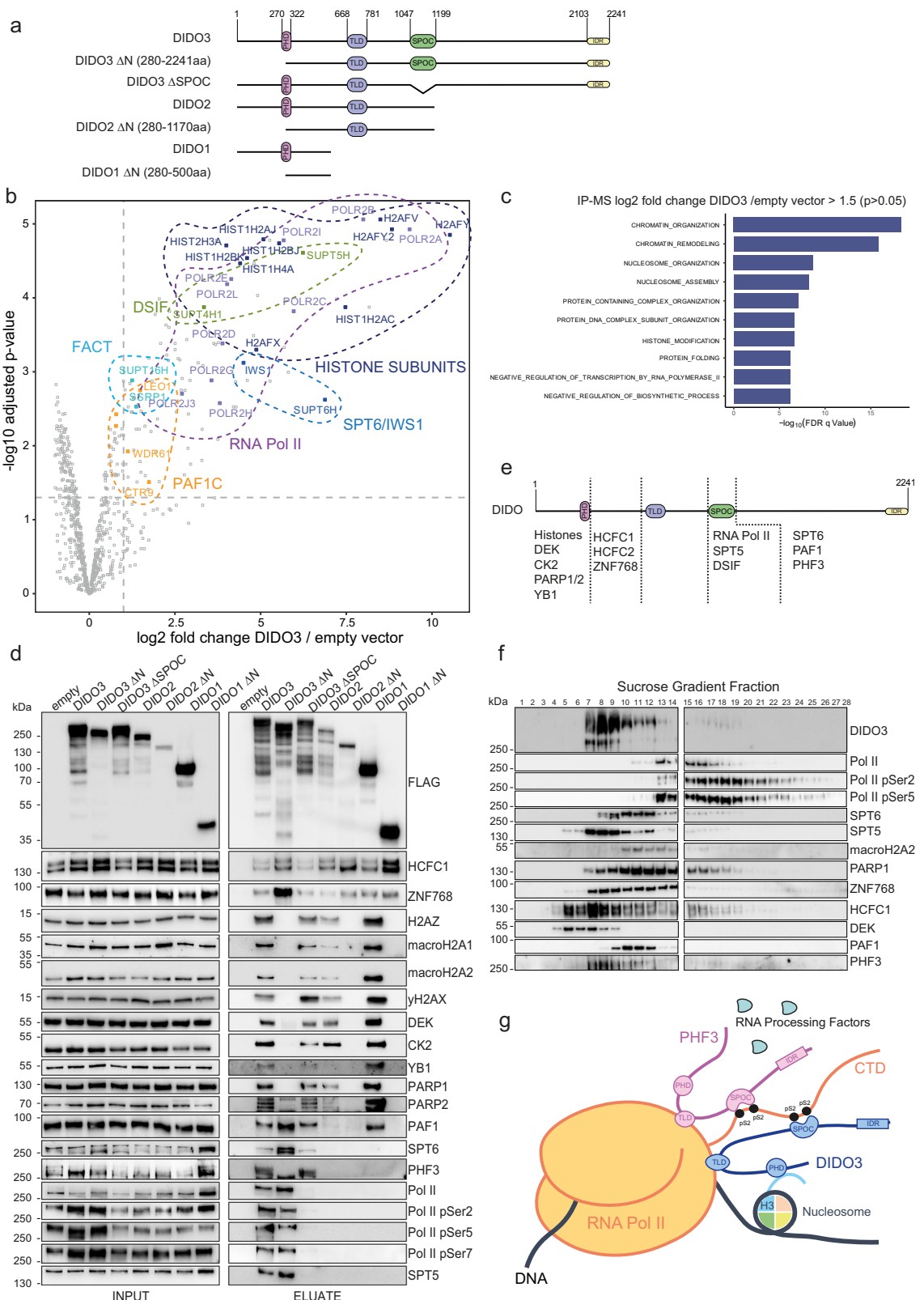

CTD[7]. Interestingly, DIDO3 interaction with transcription elongation factors SPT6 and PAF1C is neither N-terminus nor SPOC-dependent, suggesting alternative contact points through the DIDO3 C-terminal region or multivalent interactions (Fig. 2d and Supplementary Fig. 3a, b). Compared to DIDO3, DIDO1 and DIDO2 interacted with chromatin but not with Pol II and the elongation factor SPT5 due to the lack of the SPOC domain (Fig. 2d, Supplementary Figs. 4, 5). Both CK2 and CDK11B

kinases interacted with the DIDO N-terminus, PP1-PNUTS interacted with the DIDO SPOC domain, whereas PP2A interacted with the DIDO N-terminus and C-terminus independent of the SPOC domain (Supplementary Figs. 3 and 4). All DIDO isoforms and truncations interacted with HCFC1, HCFC2 and ZNF768, while N-terminal truncations showed stronger binding (Fig. 2d). HCFC1 and HCFC2 function as transcriptional coregulators and are both essential for cell cycle

**Fig. 2 | DIDO3 interacts with chromatin and Pol II and forms a complex with PHF3. a** A schematic representation of FLAG-DIDO constructs. **b** Volcano plot for FLAG-DIDO3 IP-MS vs empty vector. The experiment was carried out in three individual replicates. Statistical calculations were performed by two-sided t-statistics using the LIMMA package in R[42]. Adjusted p-values were calculated using the Benjamini-Hochberg correction for multiple testing. Mass Spectrometry data are provided in Supplementary Data 1. **c** GO analysis of the interactome of DIDO3 (p-value < 0.05; fold change to empty vector>1.5). GSEA biological processes tool was used[47]. **d** Anti-FLAG co-immunoprecipitation followed by western blotting using antibodies against mass spectrometry hits. The experiment was performed once. **e** A schematic overview of DIDO regions bound by different interacting partners. **f** Sucrose gradient ultracentrifugation analysis of DIDO3 macromolecular complexes. Whole-cell lysates from wild-type HEK293T cells were separated on a 15-40% sucrose gradient. Fractions were checked by western blot using antibodies against DIDO3 interactors. The experiment was performed once. **g** Schematic depiction of common and specific interaction partners of DIDO3 and PHF3. Source data are provided as a Source Data file.

progression and cell proliferation[20,21]. ZNF768 is a transcription factor that promotes cell proliferation and malignant transformation[22,23]. An overview of DIDO interactors and which regions of the protein they interact with is shown in Fig. 2e.

Comparison of PHF3 and DIDO3 interactomes revealed Pol II elongation complex as a common interaction platform of these two paralogues. To assess the association of PHF3 and DIDO3 with elongating Pol II in cells, we performed high-resolution Airy scan imaging measuring the degree of colocalization between endogenously mEGFP-tagged PHF3 or DIDO3 and Pol II phosphorylated at Ser2 of the CTD. We used a previously generated PHF3-mEGFP cell line[3] and employed CRISPR/Cas9 to endogenously tag DIDO3 with mEGFP in WT, PHF3 KO and PHF3 ΔSPOC background (Supplementary Fig. 6) and to knock-out all DIDO isoforms in PHF3-mEGFP background (Supplementary Fig. 8e, f and Supplementary Fig. 18d). High-resolution Airy scan imaging revealed stronger association of Pol II with DIDO3 than PHF3 (Supplementary Fig. 7a, b). While PHF3 SPOC showed 10-fold higher affinity for Pol II pSer2 CTD compared to DIDO3 SPOC[7], additional contact points between DIDO3 and Pol II may confer stronger binding. DIDO3 colocalization with Pol II was reduced upon inhibition of pause release with a specific CDK9 inhibitor NVP-2 or splicing inhibition with pladienolide B (Pla-B) (Supplementary Fig. 7c–f). DIDO3 loss or PHF3 loss/PHF3 SPOC deletion did not affect PHF3 or DIDO3 colocalization with Pol II respectively (Supplementary Fig. 8).

DIDO3 was found to interact strongly with histones and chromatin-associated factors through its N-terminal region including the PHD domain, while PHF3 does not bind histones but interacts with RNA processing factors. Mass spectrometry analysis of differential Pol II interactome in PHF3 KO vs WT cells revealed reduced binding of different RNA processing factors to Pol II, including splicing factors and nuclear speckle proteins (Supplementary Fig. 9a, Supplementary Data 2). In DIDO KO cells we observed reduced binding of chromatin factors to Pol II (Supplementary Fig. 9b, Supplementary Data 3). Overall, these results suggest that DIDO and PHF3 interact with and promote the association of chromatin and RNA processing factors to Pol II respectively.

## DIDO3 and PHF3 form a macromolecular complex

DIDO3 interaction with PHF3 was neither dependent on the DIDO3 SPOC domain nor its PHD domain, but was absent for the shorter isoforms DIDO1 and DIDO2, suggesting that, as in the case of PAF1C and SPT6, DIDO3 and PHF3 interact through the C-terminal region or have multiple contact points (Fig. 2d, e). Sucrose gradient ultracentrifugation revealed complex formation between DIDO3 and PHF3, whereby both proteins appeared in the same fractions (Fig. 2f). We could identify two subcomplexes: a smaller molecular weight complex with a higher abundance of DIDO3 and PHF3 and a larger molecular weight Pol II complex where DIDO3 and PHF3 were less abundant (Fig. 2f). The smaller molecular weight complex consisted of Pol II-associated factors such as SPT5, SPT6 and PAF1, chromatin modulators such as PARP1 and DEK, and histones such as macroH2A (Fig. 2f). To examine complex formation in intact cells, we generated a CRISPR/Cas9 cell line with endogenously labeled PHF3-mScarlet and DIDO3-mEGFP (Supplementary Fig. 10). High-resolution Airy scan imaging

revealed a high degree of colocalization between PHF3 and DIDO3 (Supplementary Fig. 11a, b). Moreover, PHF3 and DIDO3 seem to mutually affect protein stability: DIDO3 levels were reduced in PHF3 ΔSPOC and KO cells compared to WT cells due to increased degradation, whereas PHF3 levels were increased in DIDO3 KO cells (Supplementary Fig. 11c–f).

Overall, our results show that DIDO3 and PHF3 form a complex that spans chromatin and Pol II: DIDO3 anchors the complex to chromatin while PHF3 provides the connection to RNA processing factors (Fig. 2g).

## Intrinsically disordered C-terminal regions of DIDO3 and PHF3 form condensates that sequester Pol II and regulate transcriptional output

The C-terminus of PHF3 and DIDO3 contains a low complexity domain characterized by a high PScore as a measure of π−π interactions and a high catGRANULE score, both indicating propensity for liquid-liquid phase separation (Fig. 3a). We previously showed that PHF3 sequesters pSer2 Pol II within liquid-like condensates in vitro and colocalizes within Pol II clusters in cells[3]. We tested whether C-terminal regions of PHF3 and DIDO3 can form condensates in vitro (Supplementary Fig. 12). PHF3 1595-2039aa and DIDO3 2085-2240aa showed most prominent condensation in vitro, at a physiological salt concentration (150 mM) and without a crowding agent, and will be referred to as PHF3 IDR and DIDO3 IDR (Fig. 3b, c and Supplementary Fig. 12). DIDO3 IDR shows properties of a mixed-charge domain, which is interspersed with arginines and aspartates and partially structured based on the AlphaFold prediction (α-helix 2113-2195aa). PHF3 IDR comprises hydrophobic and charged clusters. The median condensate area was larger for DIDO3 IDR ($4.3\,\mu m^2$) compared to PHF3 IDR ($2.2\,\mu m^2$) (Fig. 3c). DIDO3 and PHF3 IDRs colocalized within condensates and drove partitioning of Pol II within the condensates (Fig. 3b, c). Co-mixing of DIDO3 IDR and Pol II gave rise to larger condensates (median area $7\,\mu m^2$), which was not the case for PHF3 IDR (Fig. 3b,c).

Co-immunoprecipitation experiments revealed that PHF3 IDR does not interact with Pol II, whereas DIDO3 IDR strongly interacts with Pol II (Fig. 3d,e) and thereby potentiates mixed condensate formation (Fig. 3b,c). The removal of DIDO3 IDR did not impair its interaction with Pol II, which is mediated through the SPOC domain, but reduced its interaction with PHF3, PARP1 and PAF1 (Fig. 3f).

To assess the contribution of the IDRs and the DIDO SPOC domain to DIDO3 and PHF3 cellular localization, we used CRISPR/Cas9 to generate endogenously tagged DIDO3 ΔIDR-mEGFP, DIDO3 ΔSPOC-mEGFP and PHF3 ΔIDR-mScarlet cell lines (Supplementary Figs. 6, 13, 14). Endogenously tagged DIDO3-mEGFP and PHF3-mScarlet in the previously generated double-tagged cell line (Supplementary Fig. 10) showed uneven distribution throughout chromatin, with highly compacted areas showing slightly weaker DIDO3 signal, suggesting that they may preferentially localize to euchromatin (Fig. 3g and Supplementary Fig. 15a–d). Overall, ΔIDR mutants showed minor differences in distribution and cluster size (Fig. 3g). However, high-resolution Airy scan imaging revealed reduced colocalization of DIDO3 ΔIDR with Pol II pSer2, which was comparable to DIDO3 ΔSPOC (Fig. 4a, b). Likewise, PHF3 ΔIDR showed reduced colocalization with Pol II pSer2 (Fig. 4c, d). This suggests that DIDO3 and PHF3 IDRs facilitate clustering with Pol II.

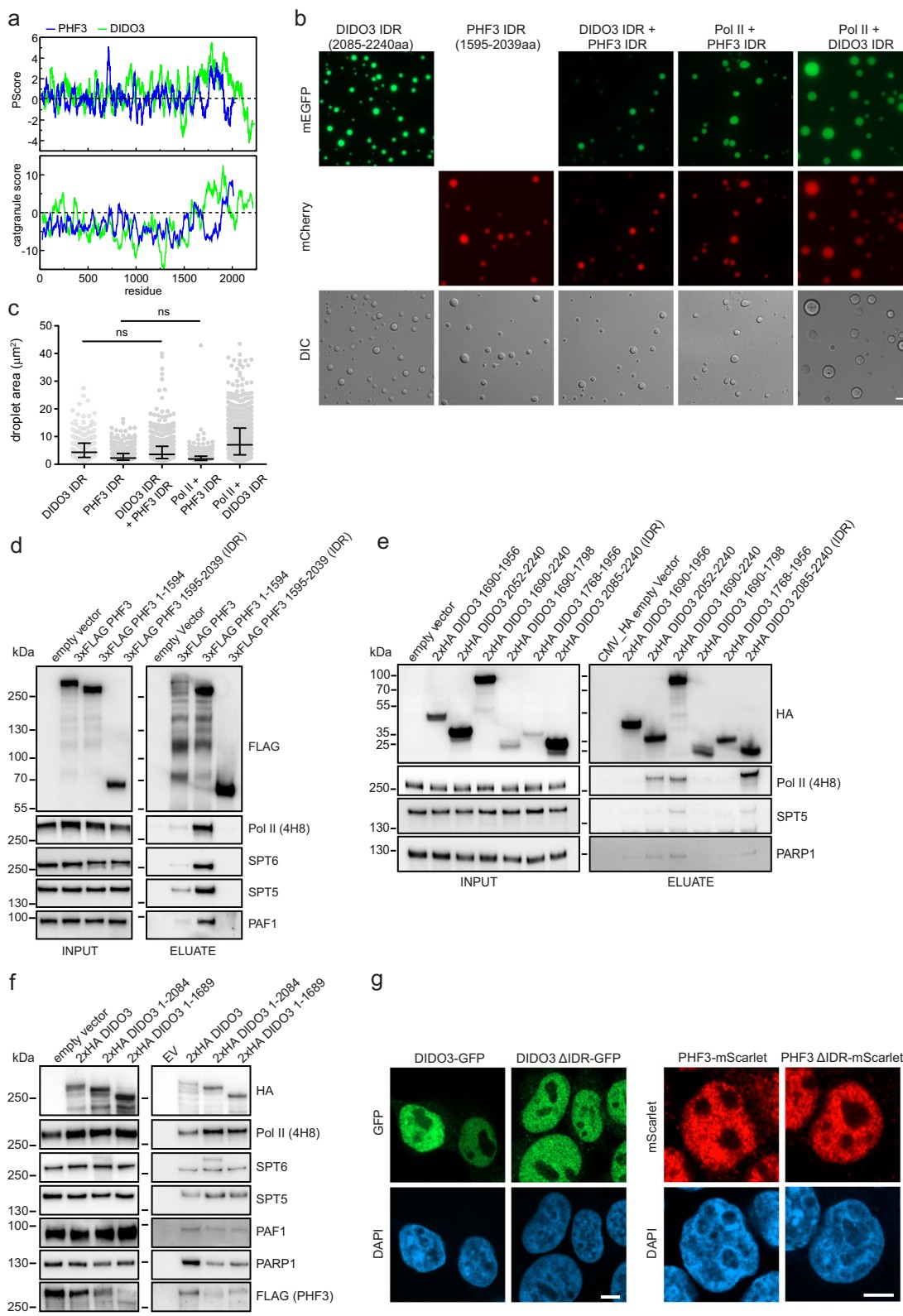

LLPS has been implicated in transcription regulation as a means of locally increasing concentration of different factors that drive a particular step during the transcription cycle. For example, LLPS of unphosphorylated CTD triggers Pol II clustering with the Mediator complex[24], whereas phosphorylated CTD clusters with RNA processing factors[25]. Shortening CTD length was shown to reduce transcription[26], whereas removal of TF IDRs has neutral or slightly inhibitory effect on

transcription[27]. To examine whether DIDO3 and PHF3 IDR-mediated Pol II clustering has an impact on transcription regulation, we performed RNA-seq analysis showing that both IDR deletions result in transcriptional deregulation (Fig. 4e, f and Supplementary Data 4). DIDO ΔIDR showed upregulation of 118 transcripts and downregulation of 75 transcripts (fold change>2, p-value < 0.05) (Fig. 4e). In contrast, PHF3 ΔIDR showed downregulation of 3891 transcripts with

**Fig. 3 | DIDO3 and PHF3 C-terminal regions form condensates. a** PScore and catGRANULE score for PHF3 and DIDO3. **b** Representative images of in vitro LLPS assays with 25 μM DIDO IDR or PHF3 IDR and 1.5 μM Alexa488-Pol II. Scale bar=5 μm. **c** Quantification of condensate area (μm²). N = 262 (DIDO IDR); 460 (PHF3 IDR); 582 (DIDO IDR + PHF3 IDR); 378 (Pol II + PHF3 IDR); 560 (Pol II + DIDO IDR). The experiment was performed in two independent replicates. Data are presented as median (central line) with 25-75% interquartile range (error bars). One-way ANOVA with Tukey's multiple comparison test was performed to determine p-values. All p-values are <0.0001 except for the indicated pairs (ns=non-significant). **d** Anti-

FLAG co-immunoprecipitation of 3xFLAG PHF3 full-length and truncations. PHF3 IDR does not interact with Pol II. **e,f** Anti-HA co-immunoprecipitation of 2xHA DIDO3 **e** C-terminal constructs **f** full-length and C-terminal truncations. DIDO3 IDR interacts with Pol II. Experiments in **d-f** were performed once. **g** Immunofluorescence analysis of mEGFP-tagged DIDO3 and mScarlet-tagged PHF3 cell lines using an anti-GFP antibody and an anti-FLAG antibody to enhance the endogenous Scarlet signal. Scale bar=5 μm. Experiments were performed in two independent replicates, representative images are shown. Source data are provided as a Source Data file.

---

the top downregulated genes being implicated in neurogenesis (Fig. 4f and Supplementary Fig. 15e). Interestingly, H3K9me3 showed altered distribution in PHF3 ΔIDR cells, but not in DIDO3 ΔIDR, with clearly defined clusters resembling mouse chromocenters, suggesting that loss of PHF3 IDR may result in genome reorganization and increased heterochromatin formation (Fig. 4g, h). Enhanced H3K9me3 clustering in PHF3 ΔIDR cells is concordant with gene downregulation. In conclusion, our data suggest that DIDO and PHF3 IDRs contribute to gene expression regulation by mediating clustering of the Pol II transcription machinery and in the case of PHF3 IDR possibly also by modulating genome organization.

## Genome-wide transcriptional changes caused by DIDO perturbation

Our data show that PHF3 and DIDO form a complex with Pol II and regulate Pol II clustering (Figs. 2–4). Furthermore, loss of PHF3 and its SPOC domain trigger upregulation of DIDO3 that contains the SPOC domain (Fig. 1). Thus we hypothesized that DIDO3 and PHF3 coregulate gene expression. To explore the functional connection between PHF3 and DIDO3, we used CRISPR/Cas9 to generate DIDO KO HEK293T cell lines by targeting exon 3, which was expected to result in a complete protein knock-out, and exon 7, which retains DIDO1 and DIDO2 isoforms, but not DIDO3 (Supplementary Fig. 16a). The knockouts were generated in PHF3 WT, KO and ΔSPOC backgrounds. The strategy to generate a full DIDO KO with a gRNA targeting the very beginning of exon 3 showed retention of a slightly smaller DIDO3 band (Supplementary Fig. 16b). This led us to examine whether the *DIDO1* gene might contain alternative transcriptional start sites downstream of the start codon, which, when combined with an alternative translational start codon, would produce a shorter version of the proteins. CAGE data showed that there are in fact three transcriptional start sites for the *DIDO1* gene (Supplementary Fig. 17), but all are upstream of exon 3, indicating that the shorter versions of the DIDO isoforms can only be caused by ribosomal leaky scanning. We identified two alternative start codons in exon 3 and exon 4 (ATG2 and ATG3 in Supplementary Fig. 16a). A gRNA targeting exon 4 resulted in a complete DIDO knock-out based on western blot and immunofluorescence analysis (Supplementary Fig. 16b, d)[7]. The partial KO generated by targeting exon 3 expresses a DIDO3 isoform that lacks the first 88 amino acids and will be referred to as DIDO N[1–88]-Isoform KO. A gRNA targeting exon 7 was used to generate DIDO Long Isoform KO (Supplementary Fig. 16b,d), which retains the short isoform DIDO1 (Supplementary Fig. 16c, 18). To elucidate the importance of the DIDO3 SPOC domain, we used CRISPR/Cas9 to remove this domain in WT, PHF3 KO and PHF3 ΔSPOC background (Supplementary Fig. 14a, 16b, 19a). Western blot analysis showed additional degradation products for DIDO ΔSPOC, suggesting that removal of the SPOC domain may impair protein stability as in the case of PHF3[3] (Supplementary Fig. 16b). An overview of all cell lines is shown in Supplementary Fig. 19. Overall, our results show that complete DIDO KO is viable in HEK293T cells, in contrast to lethality shown for ESCs and reported for human cell lines[14,15].

To address the effect of different DIDO mutant cell lines on gene expression, we performed RNA-seq analysis with Drosophila spike-in normalization and determined the number and gene ontology

enrichment for genes that showed >2-fold deregulation with *p*-value < 0.05 (Fig. 5, Supplementary Figs. 20–22 and Supplementary Data 4). DIDO N[1–88]-Isoform KO showed very little transcriptional perturbation (2 genes UP, 3 genes DOWN), suggesting that the first 88 amino acids are not essential for DIDO function (Fig. 5g, Supplementary Fig. 20a). Conversely, DIDO Long Isoform KO, DIDO full KO and DIDO ΔSPOC showed a stronger phenotype with 196, 338 and 470 upregulated and 2264, 984 and 311 downregulated genes respectively, with 97 and 98 genes being upregulated or downregulated in all three mutants respectively (Fig. 5a, d, g and Supplementary Fig. 20b, e). Gene ontology analysis revealed enrichment of 'Lipid metabolic process' among upregulated genes in all three cell lines (Fig. 5h and Supplementary Fig. 21). DIDO Long Isoform KO showed upregulation of predominantly metabolic genes, suggesting that long DIDO isoforms negatively regulate the expression of metabolic genes (Supplementary Fig. 21). 'Neurogenesis' and 'Morphogenesis' categories were enriched among downregulated genes in DIDO Long Isoform KO, DIDO full KO and DIDO ΔSPOC, while 'Proliferation' genes were also downregulated in DIDO full KO and DIDO ΔSPOC (Fig. 5h and Supplementary Fig. 21). 'Cell projection organization' and 'Cell adhesion' were the most highly represented categories among upregulated genes in DIDO ΔSPOC (Fig. 5h and Supplementary Fig. 21). In comparison, RNA-seq analysis of DIDO C-terminal truncation (ΔE16) in mESCs revealed upregulation of 86 genes and downregulation of 21 genes (>2-fold deregulation, no spike-in normalization) with enrichment of genes involved in proliferation, differentiation, RNA metabolism and nervous system development[13].

## DIDO regulates cell proliferation

Genes required for proliferation were downregulated in DIDO mutants (Fig. 5h and Supplementary Fig. 21), suggesting that DIDO regulates cell proliferation. All DIDO mutant cell lines except for DIDO N[1–88]-Isoform KO showed a growth defect during a 4-day measurement of their growth rates, which was not caused by increased apoptosis (Fig. 6a and Supplementary Fig. 16e). Interestingly, DIDO ΔSPOC and DIDO ΔIDR showed more severe proliferation defects compared to the two KO cell lines (Fig. 6a,b). These two cell lines also showed decreased S and increased G2/M population according to FACS analysis (Fig. 6d and Supplementary Fig. 23). PHF3 deletion or overexpression had no effect on cell growth (Fig. 6c), whereas combined perturbation of PHF3 and DIDO showed the strongest growth defect, particularly when DIDO Full KO was combined with PHF3 KO (Fig. 6b). These results show that DIDO-mediated regulation of cell proliferation is independent of PHF3. Among the downregulated genes in DIDO mutants was the transcription factor E2F1 (Fig. 6e), which controls the expression of genes involved in proliferation[28] and may be responsible for the observed proliferation phenotypes.

## Regulation of neuronal gene expression by DIDO3 and PHF3

To further examine the crosstalk between DIDO and PHF3, we analysed differential gene expression in single and double mutant cell lines in which DIDO full KO or ΔSPOC were combined with PHF3 KO or ΔSPOC respectively (Fig. 5c, f, g, Supplementary Fig. 20c–f and Supplementary Fig. 22). We observed 462 upregulated and 295 downregulated genes in the PHF3 KO DIDO full KO double knock-out cell line (fold change

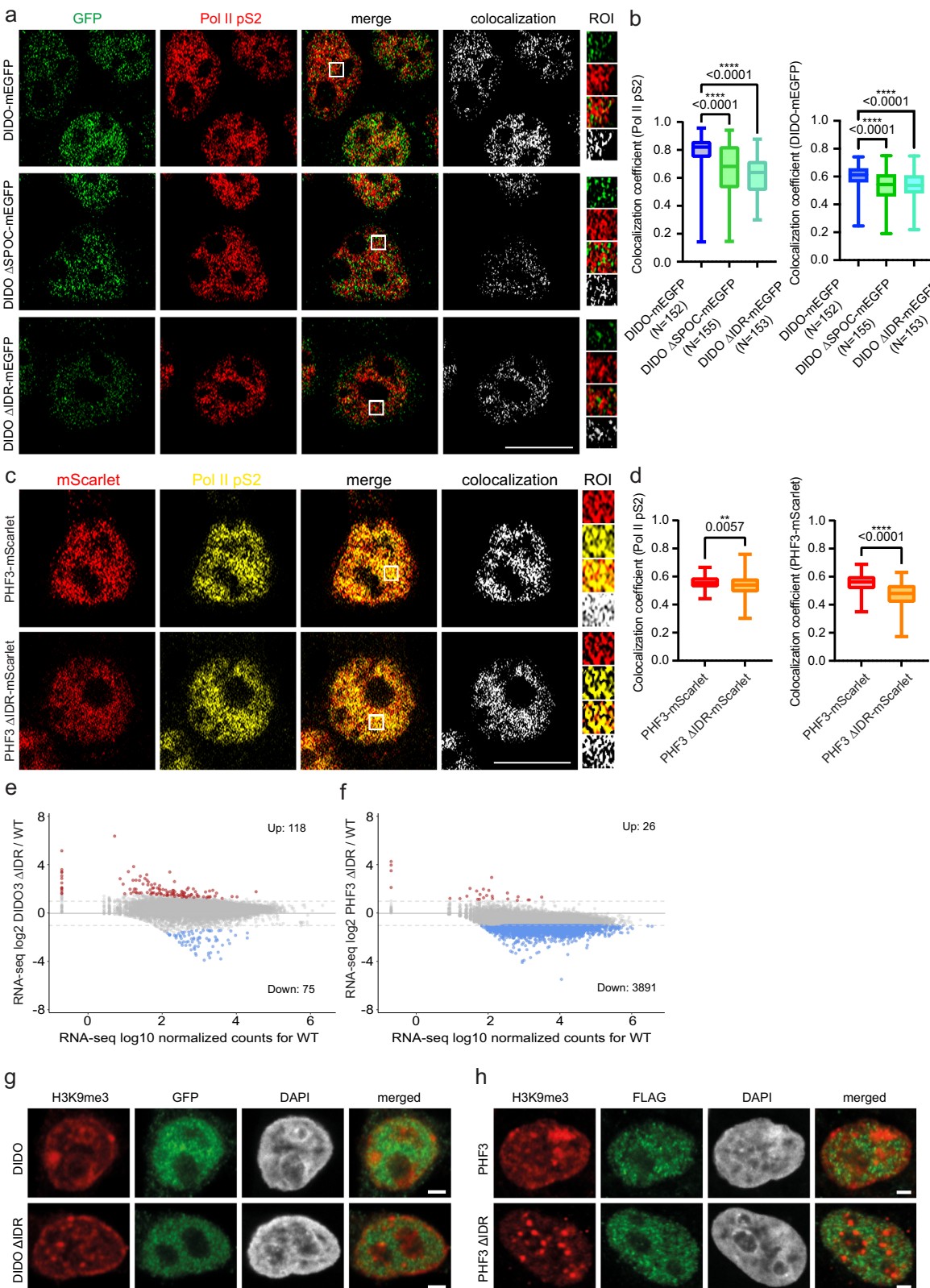

>2, p-value < 0.05). Among these, 166 upregulated and 66 downregulated genes overlapped with the DIDO full KO and 94 upregulated and 24 downregulated genes overlapped with the PHF3 KO single knock-out cell lines. 24 and 8 genes were significantly up- and downregulated in all three cell lines respectively (Fig. 5c, g and Supplementary Fig. 20e). Genes that were up- or downregulated in the double KO cell line show similar deregulation in the single KO cell lines, but

with lower fold changes, often not reaching our threshold (fold change >2, p-value < 0.05) (Fig. 7a, b). This suggests that one paralogue can partially compensate for the loss of the other. The extent of deregulation is generally larger in the DIDO full KO compared to the PHF3 KO cell line (Fig. 7a, b), suggesting that DIDO can compensate better for the loss of PHF3, possibly due to upregulation of the SPOC-containing DIDO3 isoform (Fig. 1b). Simultaneous deletion of the SPOC

**Fig. 4 | DIDO3 and PHF3 C-terminal regions facilitate clustering with Pol II and regulate transcription. a,c** Representative Airyscan high resolution images of **a** mEGFP-tagged DIDO3, DIDO ΔSPOC and DIDO ΔIDR (IF staining with rabbit anti-GFP + Alexa Fluor 488, green) and Pol II pS2 (Alexa Fluor 594, red) and **c** PHF3-mScarlet or PHF3 ΔIDR-mScarlet (IF staining with mouse anti-FLAG + Alexa Fluor 568, red) and Pol II pS2 (Alexa Fluor 647, yellow). Colocalization analysis of clusters that overlap in both channels (white). Scale bar=10 μm. **b,d** Quantification of the fraction of Pol II pS2 colocalizing with **b** DIDO or **d** PHF3 (Manders coefficient 1; left panel) or fraction of DIDO or PHF3 colocalizing with Pol II pS2 (Manders coefficient 2; right panel). Box and whiskers plot depicting the median (line), 25-75% inter-quartile range (box borders) and minimum/maximum (whiskers) are shown. Each experiment was repeated three times with comparable results. **b** One-way ANOVA with Brown-Forsythe & Welch's correction was used to determine statistical significance. **d** Two-tailed unpaired Student's t-test with Welch's correction was used to determine statistical significance. **e,f** RNA-seq analysis shows upregulation of 118 genes in DIDO3 ΔIDR and 26 genes in PHF3 ΔIDR (red dots, fold-change>2, p < 0.05) and downregulation of 75 genes in DIDO3 ΔIDR and 3891 genes in PHF3 ΔIDR (blue dots, fold-change>2, p < 0.05) compared to WT. The experiments were performed in three independent replicates. Statistical analysis was performed using two-sided Wald test as implemented in DESeq2[39]. Drosophila S2 cells were used for spike-in normalization. **g,h** Immunofluorescence images showing H3K9me3 in **g** mEGFP-tagged DIDO3 and DIDO ΔIDR, and **h** PHF3-mScarlet and PHF3 ΔIDR-mScarlet. Scale bar=2 μm. Experiments were performed twice, representative images are shown. Source data are provided as a Source Data file.

domains of DIDO and PHF3 resulted in a global downregulation of 7679 genes, with enrichment of genes required for 'Neurogenesis' and 'Morphogenesis' among the 500 most downregulated genes (Fig. 5f, g and Supplementary Fig. 22b). The double mutant displays a much more severe phenotype than individual SPOC deletions (Fig. 5d–g), indicating that the presence of both SPOC-deficient protein variants may have a dominant negative effect on gene expression.

In addition to cooperative regulation, DIDO and PHF3 showed antagonistic effects in gene expression regulation, which were most prominent for neuronal genes with DIDO acting mainly as a positive regulator and PHF3 as a negative regulator (Fig. 5h). Overall, we categorized DIDO3-PHF3 crosstalk as: 1) negative co-regulators (genes that are upregulated in DIDO KO/ΔSPOC, PHF3 KO/ΔSPOC and 2xKO/ΔSPOC); 2) positive co-regulators (genes that are downregulated in DIDO KO/ΔSPOC, PHF3 KO/ΔSPOC and 2xKO/ΔSPOC); 3) antagonistic co-regulators (genes that are downregulated in PHF3 KO/ΔSPOC and upregulated in DIDO KO/ΔSPOC or vice versa) (Fig. 5h, Fig. 7). We validated these findings by RT-qPCR for a representative gene in each category (Fig. 7c–f).

### DIDO and PHF3 regulate neuronal differentiation of mESCs

We previously found that during neuronal differentiation of mESCs PHF3 ensures timely expression of several key neuronal factors that regulate neuronal cell fate including Ascl1, Nestin, Pou3f2 and Sox21[3]. In mESCs, we could generate a complete Phf3 KO[3] but the complete Dido KO is not viable, which is why we used a heterozygous Dido KO[29] (Fig. 8a, b). Phf3 KO mice generated by the International Mouse Phenotyping Consortium (IMPC) exhibit neuronal dysfunction in the form of impaired auditory brainstem response and impaired startle reflex (www.mousephenotype.org), showing that Phf3 loss also interferes with neuronal development in vivo. DIDO is essential for mouse development, while heterozygous DIDO3 truncations cause perinatal mortality featuring craniofacial and neurobehavioral abnormalities[17]. Our in vitro results corroborate these findings by showing that Dido, like Phf3, is essential for neuronal differentiation (Fig. 8c, d) and that heterozygous Dido depletion also causes premature derepression of key neuronal transcription factors such as Ascl1, Nestin, Pou3f2 and Sox21 (Fig. 8e). Dido and Phf3 KO mESCs did not show major changes in cell cycle profiles compared to WT (Supplementary Fig. 24) and retained differentiation potential up to the neural stem cell (NSC) stage, as they were able to differentiate into astrocytes (Fig. 8c). Dido expression levels were increased in NSCs and neurons compared to stem cells, as previously reported for Phf3, and Dido was upregulated in Phf3 KO and vice versa (Fig. 8f). Overall, our data suggest that Dido and Phf3 promote neuronal fate specification by regulating the timing of the expression of neuronal transcription factors as stem cells are committed to neural fate.

## Discussion

Paralogous proteins have evolved by gene duplication and may exhibit distinct tissue expression patterns and genomic binding. DIDO3 and PHF3 paralogues show similar gene expression levels and tissue expression patterns[30], but distinct chromatin binding: DIDO3 occupies H3K4me3 regions through its PHD whereas PHF3 PHD has lost the chromatin-binding capacity due to aromatic cage mutation[5]. Paralogues often have redundant and compensatory functions and inactivation of one paralogue was shown to elicit upregulation of the other as a back-up mechanism[31]. Consistently, we found that deletion of PHF3 results in downregulation of the smallest DIDO1 isoform and upregulation of the largest DIDO3 isoform (Fig. 1b), which may compensate for PHF3 loss through SPOC domain-mediated binding to Pol II CTD. To our knowledge, this is the first example of paralogue buffering through isoform switching rather than a change in overall gene expression levels. In accordance with these results, genes that are up- or downregulated in the PHF3 KO DIDO full KO double knock-out cell line are deregulated to a much smaller extent in the single knock-outs, in particular in PHF3 KO (Fig. 7a, b), indicating partial compensation by one paralogue for the loss of the other (Fig. 9a).

However, paralogous proteins cannot fully compensate for each other as they may have specific gene targets or specific regulatory functions on the same genes[32]. For example, DIDO positively regulates genes required for proliferation such as E2F1 and DIDO KO cells show a severe growth phenotype unlike PHF3 KO cells (Fig. 6). Moreover, DIDO negatively regulates genes required for lipid metabolism, which seems to be mainly mediated through the long DIDO3 isoform (Figs. 5h, 9b and Supplementary Fig. 21). The regulation of neuronal gene expression emerged as the common function of the two paralogues, which show both cooperative (agonistic) and antagonistic effects on these genes with PHF3 acting mainly as a negative regulator and DIDO as a positive regulator (Figs. 5h, 9c). This is consistent with in vitro and in vivo neuronal phenotypes resulting from their depletion (Fig. 8)[3,17].

Paralogous genes may provide a back-up against human mutations that would result in developmental failure and often share expression profiles across tissues[33]. Likewise, DIDO3 and PHF3 show similar expression patterns with lower expression levels in the heart, liver, pancreas, kidney and brain, and higher expression levels in the lung, skin and reproductive organs[30]. DIDO isoform switching upon PHF3 inactivation may act as a phenotypic buffering mechanism to safeguard organism development. Accordingly, Phf3 knock-out mice are viable, whereas Dido KO is embryonic lethal in mice, suggesting that DIDO has specific dominant functions such as the regulation of cell proliferation, which may involve interactions with HCFC1 and HCFC2 transcription factors (Fig. 2c).

Regulation of gene expression by DIDO and PHF3 involves multiple domains and interaction networks, which can define their specific or co-regulatory functions. DIDO interacts with chromatin and chromatin-associated factors through the PHD, while PHF3 primarily interacts with the Pol II elongation machinery through its SPOC domain and TLD, which can outcompete TFIIS from Pol II, and tethers RNA processing factors to the Pol II machinery[3,7]. The mechanisms that determine gene specificity for different regulatory categories remain to be addressed in future studies.

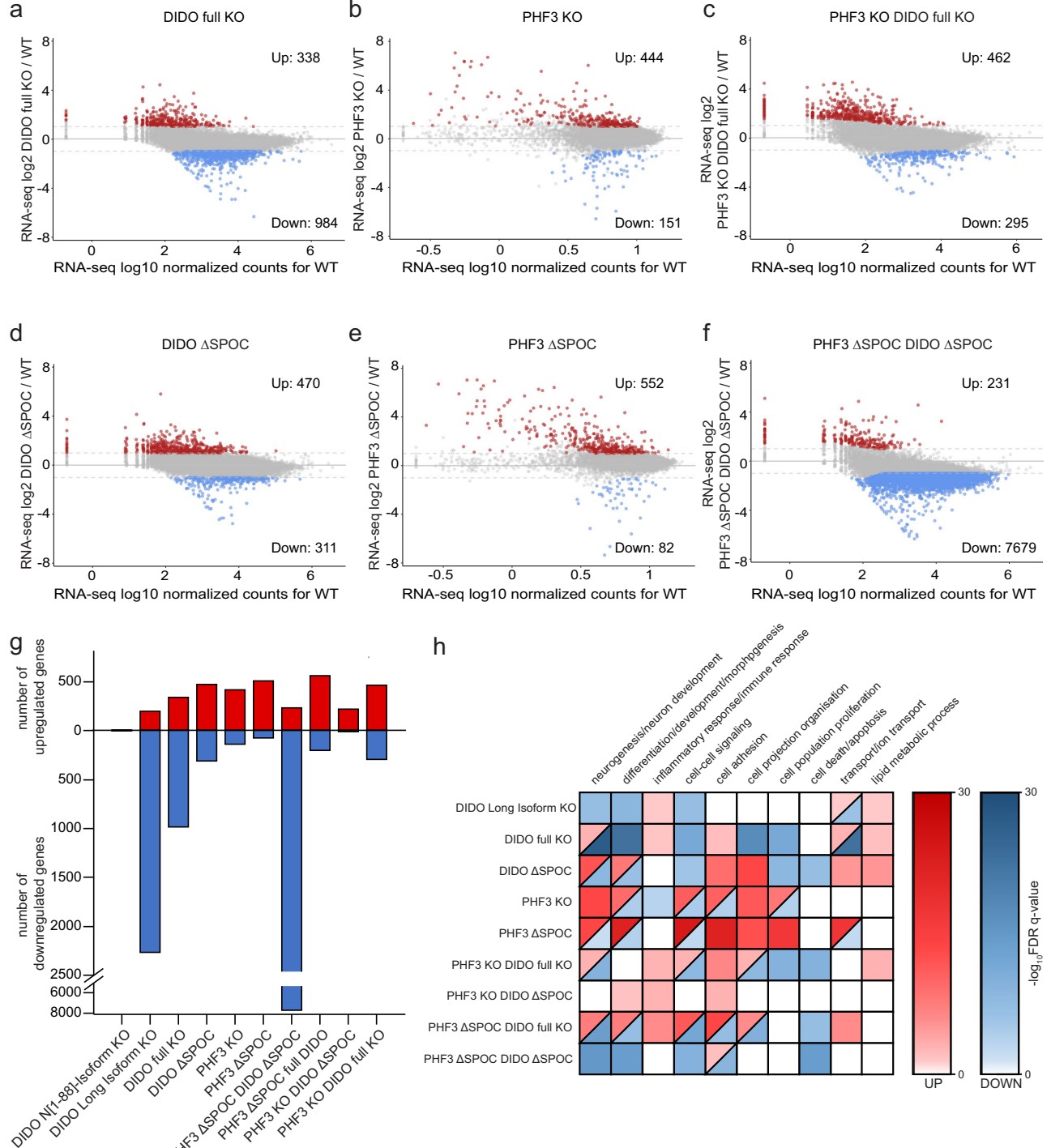

**Fig. 5 | RNA-seq analysis of single mutants and combined mutants of DIDO and PHF3 in HEK293T cell lines. a-f** MA plots showing RNA-seq log2 fold change (mutant/WT) versus log10 mean expression in WT for **a** DIDO full KO, **b** PHF3 KO, **c** PHF3 KO DIDO full KO, **d** DIDO ΔSPOC, **e** PHF3 ΔSPOC, **f** PHF3 ΔSPOC DIDO ΔSPOC. Red and blue dots indicate upregulated and downregulated genes respectively with fold-change>2, p < 0.05. The experiments were performed in three independent replicates. Statistical analysis was performed using two-sided Wald test as implemented in DESeq2[39]. Drosophila S2 cells were used for spike-in

normalization. **g** Bar chart showing the number of differentially expressed genes (fold-change>2, p < 0.05) in different genotypes. Upregulated genes are shown in red, downregulated genes in blue. **h** A heatmap of top three GO terms categories among up- or downregulated genes in at least two genotypes. White signifies no enrichment, red gradient denotes enrichment among upregulated genes and blue gradient denotes enrichment among downregulated genes. The gradient is based on the -log10 FDR q-value from the GSEA Biological processes tool[47]. Source data are provided as a Source Data file.

DIDO3 and PHF3 C-terminal IDRs can form condensates in vitro and promote association with Pol II clusters in cells, which adds another layer to their regulation of gene expression. IDR deletions showed profound effects on gene deregulation (Fig. 4e,f). A change in local clustering of the transcription machinery was previously

also shown to have profound consequences on gene expression programmes, with enhanced phase separation properties of SEC (super elongation complex) resulting in rapid transcriptional induction due to enhanced compartmentalization of the CDK9 kinase[34]. How exactly DIDO3 and PHF3 modulate the clustering of

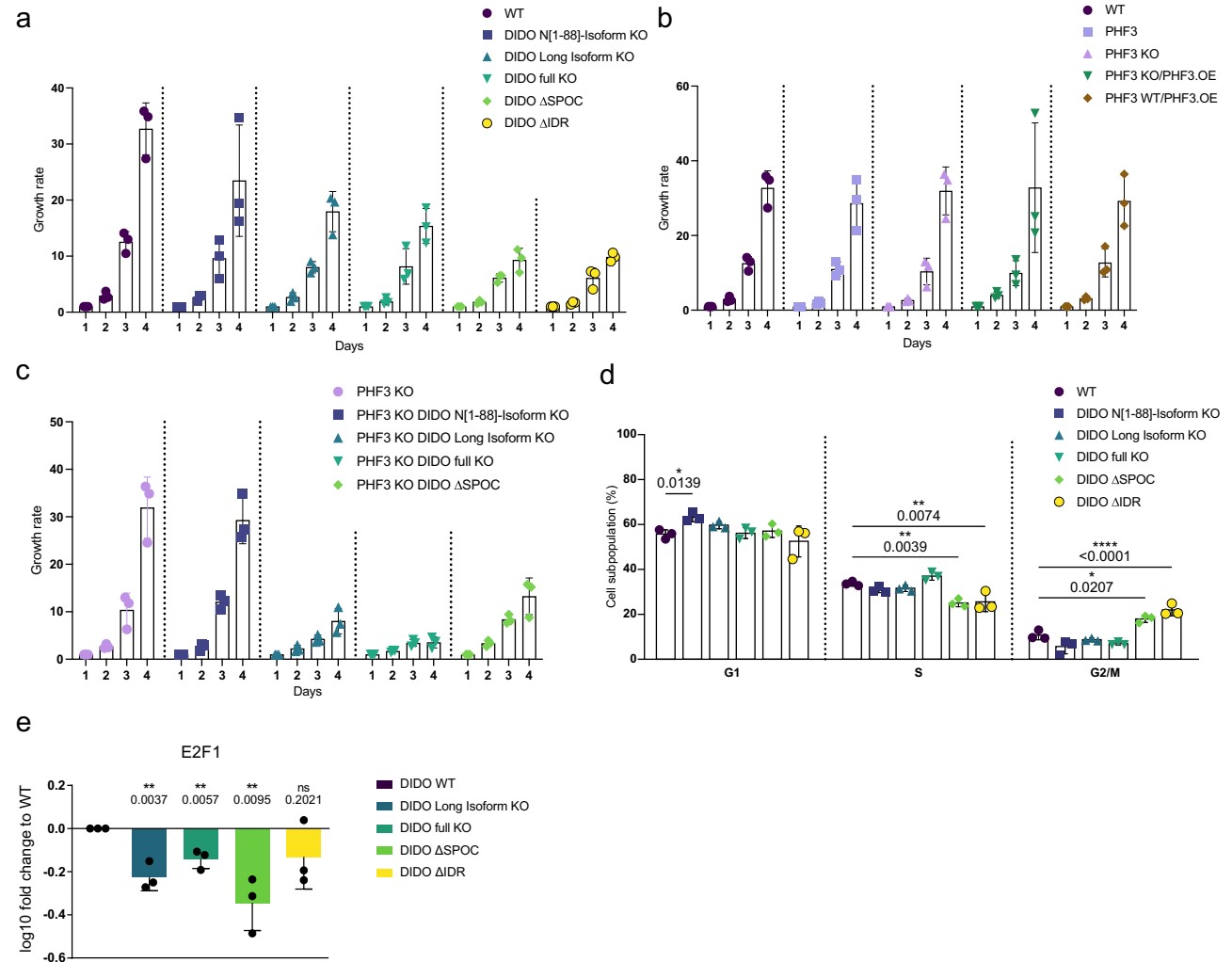

**Fig. 6 | DIDO depletion impairs cell growth. a,b,c** Growth rate of **a** DIDO mutant cell lines, **b** PHF3 mutant cell lines and **c** combined PHF3 and DIDO mutant cell lines. Growth rate was calculated by dividing the cell number at Day x by the cell number at Day 1. **d** FACS cell cycle analysis shows G1 phase cell accumulation for DIDO N[1–88]-Isoform KO and increased G2/M population for DIDO ΔSPOC and DIDO ΔIDR. Two-way ANOVA was used to determine statistical significance. **e** RT-qPCR analysis of E2F1 expression. cDNA was reverse transcribed using oligo-dT primers.

qPCR primers were designed to span exons. Gene expression was normalized to TBP and differential expression levels were expressed as a fold change compared to DIDO WT. One-tailed, two-sample equal variance t-test was used to determine p-values. Experiments in **a-e** were performed in three independent replicates. Data are presented as mean values ± standard deviation. Source data are provided as a Source Data file.

Pol II CTD and elongation machinery remains to be addressed in future studies.

In summary, we showed that DIDO and PHF3 share common gene targets and co-regulate gene expression by acting as agonists or antagonists and that DIDO3 is upregulated in PHF3 KO cells as a compensatory mechanism to buffer potentially deleterious phenotypes during development. The Pol II elongation complex is the common binding platform of DIDO3 and PHF3, which recruit chromatin modulators and RNA processing factors to Pol II and may thereby create a microenvironment comprising specific factors that fine-tune gene expression.

## Methods
### Plasmids
To generate CRISPR/Cas9 cell lines, gRNAs targeting the genomic region of interest were designed and cloned between BbsI sites into pX458 plasmid encoding Cas9 nuclease-T2A-EGFP or pX461 plasmid encoding Cas9 nickase-T2A-EGFP[35]. gRNA sequences are listed in Supplementary Data 5. Repair templates with -1kbp homology arms, CMV10 DIDO plasmids and DIDO protein expression plasmids were

generated by Gibson assembly using NEBuilder HiFi DNA assembly mix (NEB). CMV10 PHF3 plasmids and PHF3 protein expression plasmids were generated by restriction enzyme cloning. CMV10 PHF3, PHF3 ΔSPOC, PHF3 NLS-SPOC, DIDO3 and DIDO3 ΔSPOC were generated previously[7]. Cloning primers are listed in Supplementary Data 5. Plasmids are listed in Supplementary Table 2.

### Cell culture and cell line generation
All cell lines used in this study are listed in Supplementary Table 3. HEK293T cells and MEFs were grown in Dulbecco's modified Eagle's Medium (DMEM 4.5 g/L glucose) (Sigma) supplemented with 10% fetal bovine serum (Sigma), 1% L-glutamine (Sigma), 1% penicillin-streptomycin (Sigma) at 37 °C under 5% $CO_2$. mESCs were cultured on 0.2% gelatin coated plates in ES-DMEM medium supplemented with LIF and 2i. Drosophila S2 cells were grown in Schneider's Drosophila Medium (Gibco) supplemented with 10% fetal bovine serum (Sigma) at 28 °C. HEK293T PHF3 KO, PHF3 ΔSPOC, PHF3-GFP, DIDO full KO and DIDO ΔSPOC and Phf3 and Dido1 KO mESCs had been generated previously[3,7,29]. To generate HEK293T DIDO KO cell lines (N[1–88]-Isoform KO and Long Isoform KO in wildtype and PHF3 KO

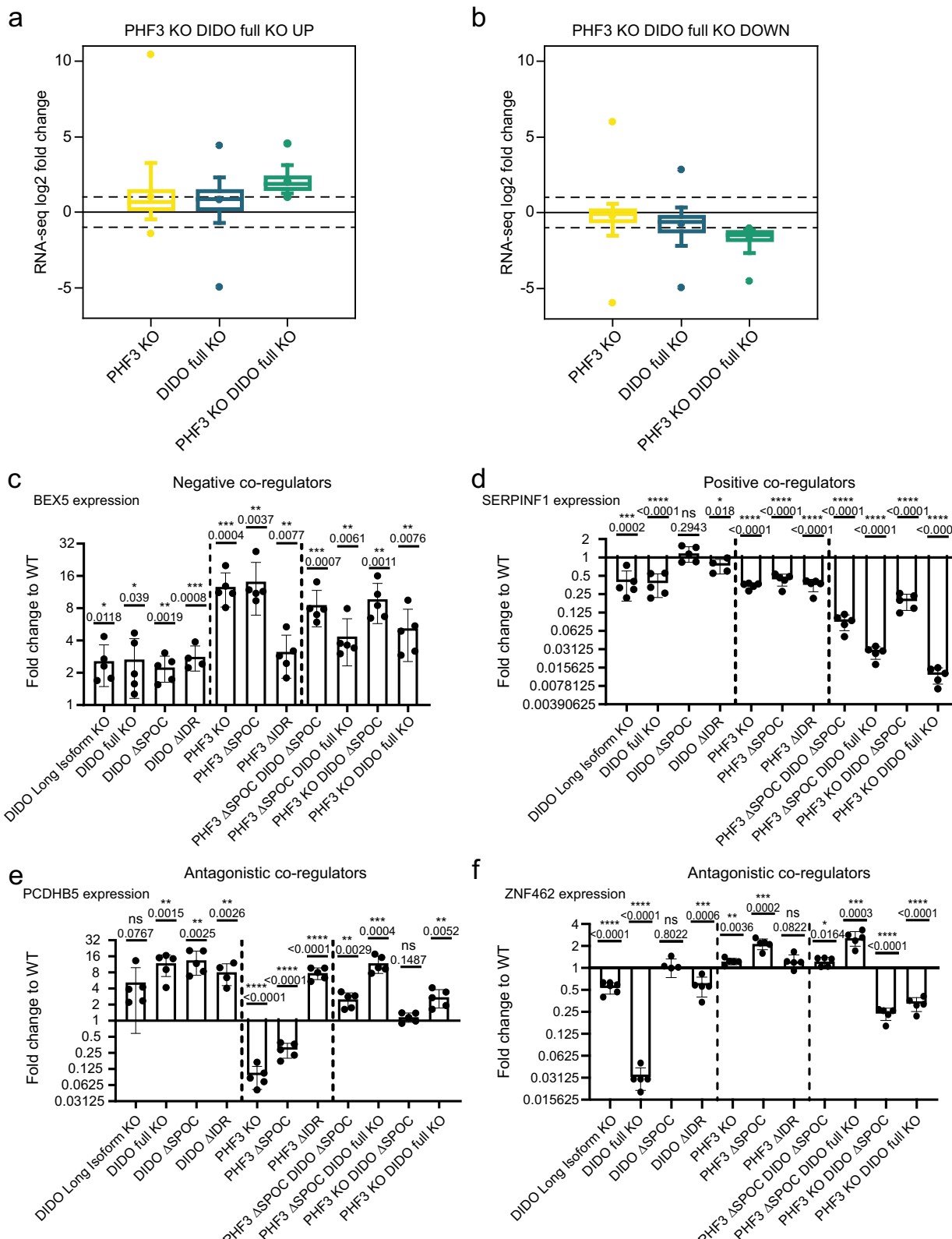

**Fig. 7 | Co-regulation of gene expression by DIDO and PHF3. a,b** Box plots showing the distribution of RNA-seq log2 fold changes derived from three biologically independent experiments in PHF3 KO, DIDO full KO and PHF3 KO DIDO full KO for genes that are **a** upregulated (n = 410 genes) or **b** downregulated (n = 287 genes) in PHF3 KO DIDO full KO. Box plots show the median (central line), the 25–75% interquartile range (box limits), and the 5-95 percentile range (whiskers). Minimum, mean and maximum values are depicted as dots. **c-f** RT-qPCR analysis of **c** BEX5 expression, **d** SERPINF1 expression, **e** PCDHB5 expression and **f** ZNF462 expression. cDNA was reverse transcribed using oligo-dT primers. qPCR primers were designed to span exons. Five biologically independent experiments were performed. Data are presented as mean values ± standard deviation. Statistical significance compared to WT is indicated. One-tailed, two-sample equal variance t-test was used to determine p-values. Source data are provided as a Source Data file.

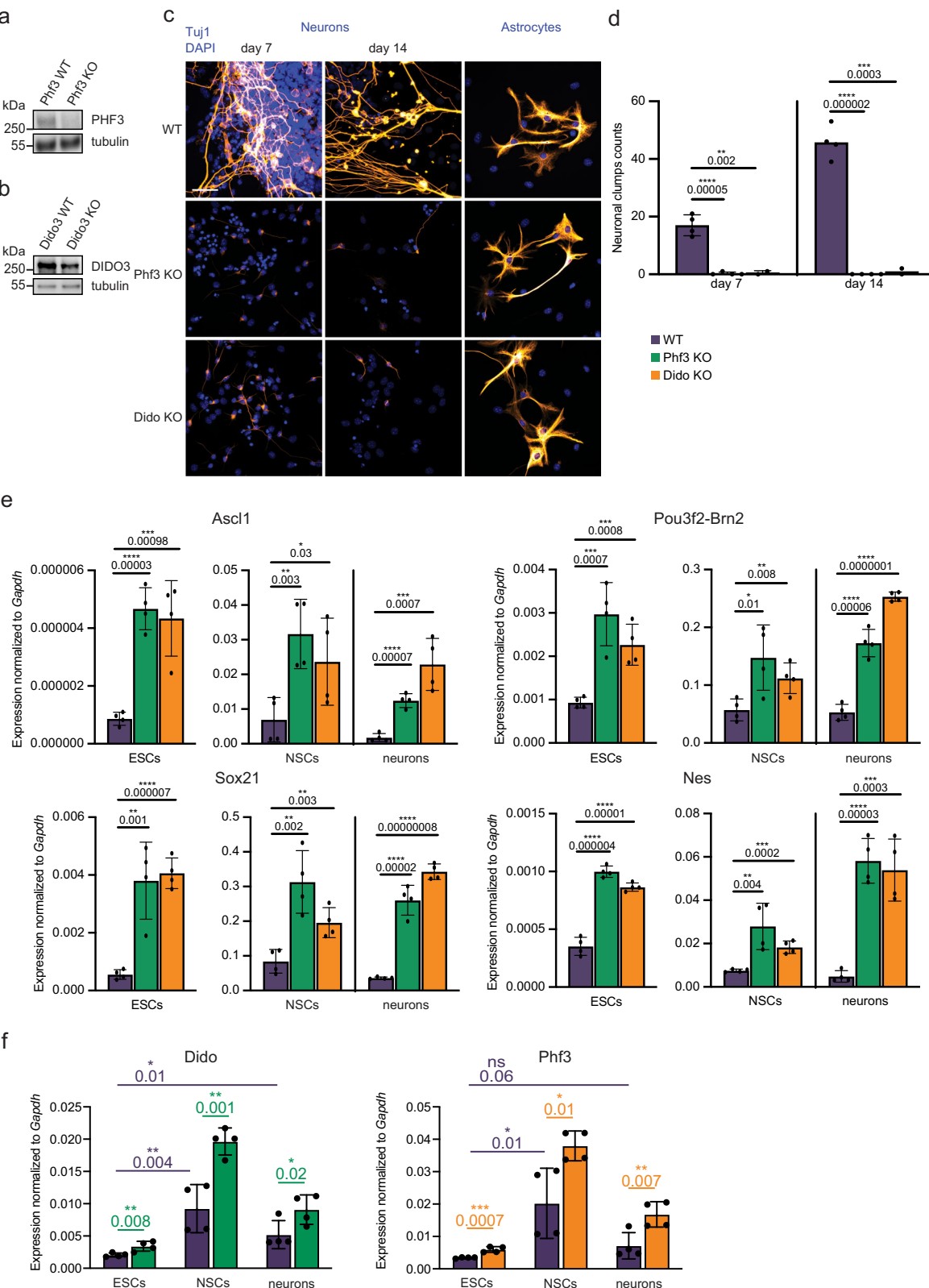

background and full KO in PHF3 KO, PHF3 ΔSPOC and PHF3-GFP background), cells were transfected with 15.6 µg pX458-gRNA plasmid in 10 cm dishes at 70% confluency using Polyethylenimine (PEI; Polysciences), followed by FACS sorting of GFP-positive cells 48-72 h after transfection. After one week, GFP-negative cells were FACS-sorted into 96-wells plates, 1 cell/well. To generate DIDO ΔSPOC cell lines in PHF3 KO or PHF3 ΔSPOC background, 1 million HEK293T PHF3 KO or PHF3

ΔSPOC cells were electroporated with 2 µg each of two pX458 plasmids encoding gRNAs targeting the genomic regions flanking the SPOC domain and 10 µg repair template consisting of either puromycin, hygromycin or blasticidin resistance cassette in antisense direction flanked by 999 bp homology arms. 72 h after electroporation 0.5 µg/mL puromycin, 100 µg/mL hygromycin or 7.5 µg/mL blasticidin was added to the culture medium. After one week, selection media was

**Fig. 8 | DIDO and PHF3 are essential for neuronal differentiation in vitro. a,b** Western blot analysis of **a** homozygous Phf3[3] and **b** heterozygous Dido KO mESCs. Tubulin was used as a loading control. Experiments were performed once. **c** Representative immunofluorescence images of WT, Phf3 KO and Dido3 heterozygous KO TuJ1-stained neurons after 7 or 14 days of neuronal differentiation and GFAP-stained astrocytes. Scale bar=40 μm. **d** Quantification of beta III tubulin (TuJ1)-positive neuronal clump formation after 7 or 14 days of neuronal differentiation. Neuronal clumps represent agglomerates of cells connected with TuJ1-positive cell projections. Four biologically independent experiments were performed. Data are presented as mean values ± standard deviation. One-tailed, two-sample equal variance t-test was used to determine p-values. **e** Comparison of expression levels of different neuronal markers in embryonic stem cells (ESCs), neuronal stem cells (NSCs) and neurons by RT-qPCR (n = 4 biologically independent experiments; mean values ± standard deviation). One-tailed, two-sample equal variance t-test was used to determine p-values. **f** RT-qPCR analysis of Dido and Phf3 expression levels in WT and KO mESCs, NSCs and neurons (n = 4 biologically independent experiments; mean values ± standard deviation). One-tailed, two-sample equal variance t-test was used to determine p-values. Source data are provided as a Source Data file.

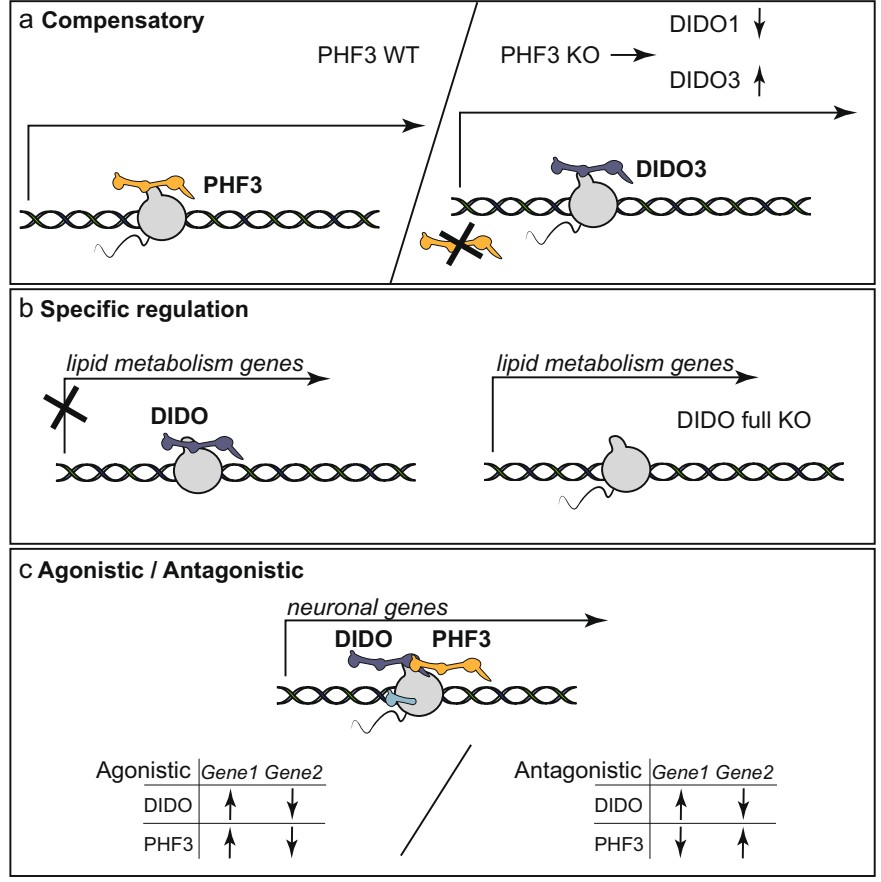

**Fig. 9 | Co-regulation of gene expression by DIDO and PHF3. a** Compensatory regulators: in the absence of PHF3, the DIDO long isoform DIDO3 is upregulated to compensate for PHF3 loss. **b** Specific regulation: DIDO negatively regulates lipid metabolism genes. **c** Agonistic regulators: DIDO and PHF3 co-regulate neuronal genes negatively or positively. Antagonistic regulators: positive regulation of neuronal genes by DIDO and negative regulation by PHF3 (prevalent) or negative regulation by PHF3 and positive regulation by DIDO.

replaced by normal culture media and cells were allowed to recover for 3 days. Cells were subsequently FACS-sorted into 96-well plates, 1 cell/well. To generate CRISPR/Cas9 endogenously EGFP-tagged DIDO3 (in wildtype, PHF3 KO, PHF3 ΔSPOC and DIDO ΔSPOC background), 1 million cells were electroporated with 2 μg pX458 plasmid encoding a gRNA targeting the DIDO3 3'end and 10 μg plasmid-borne repair template consisting of AID-EGFP-P2A-puromycin resistance cassette flanked by 999 bp homology arms. 72 h after electroporation, 0.5 μg/mL puromycin was added to the culture medium. After one week, selection media was replaced by normal culture media and cells were allowed to recover for 3 days. GFP-positive cells were subsequently FACS-sorted into 96-well plates, 1 cell/well. To generate DIDO ΔIDR (in wildtype and DIDO-GFP background), a 70% confluent 10 cm dish of WT HEK293T and DIDO-GFP cells respectively, was transfected with 7.8 μg each of two pX458 plasmids encoding gRNAs targeting the genomic regions flanking the IDR domain using PEI, followed by FACS sorting of GFP-positive cells 48-72 h after transfection. After one week,

GFP-negative cells were FACS-sorted into 96-wells plates, 1 cell/well. To generate PHF3-mScarlet (in DIDO-GFP background) and PHF3 ΔIDR-mScarlet cell lines, 70% confluent 10 cm dishes were transfected with 2 μg each of two pX461 Cas9 nickase plasmids encoding gRNAs targeting the PHF3 3'end and for PHF3 ΔIDR-mScarlet in addition 2 μg of a pX458 Cas9 nuclease plasmid encoding a gRNA targeting the region upstream of the IDR domain and 10 μg plasmid-borne repair template consisting of mAID-3xFLAG-mScarlet-P2A-hygromycin resistance cassette (PHF3-mScarlet) or mAID-3xFLAG-mScarlet-P2A-puromycin resistance cassette (PHF3 ΔIDR-mScarlet) flanked by homology arms using PEI. 72 h after transfection 0.5 μg/mL puromycin was added to the culture medium. After one week, selection medium was replaced with normal culture medium and cells were allowed to recover for 3 days. Subsequently, mScarlet-positive cells were FACS-sorted into 96-well plates, 1 cell/well. For all editing approaches, surviving clones from 96-well plates were expanded, genomic DNA was isolated using QuickExtract (Lucigen) and Cas9 target region was amplified by PCR

and successful editing was confirmed by Sanger sequencing. To generate stable cell lines, cells were transfected with CMV10 plasmids using PEI. 48 hours after transfection cells were transferred into selection media (300 µg/mL hygromycin in growth media). After two weeks, surviving cells colonies were picked using cloning cylinders. Cells were expanded and expression of integrated genes was checked using western blotting with anti-FLAG antibody.

## Proliferation assay

100,000 cells were seeded in four 6-cm dishes and cell number was counted on days 1, 2, 3, and 4 post-seeding. Cells were detached with trypsin, centrifuged at 500xg for 5 min and resuspended in 100 µL-1 mL media. Cells were counted twice using an automated cell counter (Countess II, Invitrogen, software version 1.0.249). The mean of the two cell counts was used as the final cell count. Data was analysed and plotted using GraphPad Prism (9.1.1). The data was expressed as the mean growth rate ± standard deviation of the three replicates. The growth rate was calculated as the number of cells on day 1 divided by the number of cells on the corresponding day.

## Cell lysis, SDS-PAGE and Western blotting

Cell pellets were resuspended in lysis buffer (50 mM Tris-Cl pH 8, 150 mM NlaCl, 1% Triton, 1x protease inhibitors, 1 mM PMSF, 2 mM NaF, 50 units/mL benzonase and 1 mM DTT) and lysed for 20 min on ice before centrifugation at 4 °C for 10 min. Protein concentration in the supernatant was estimated by Bradford assay. 20 µg lysate per lane were loaded onto SDS-PAGE gels, transfer onto nitrocellulose membrane was performed in transfer buffer (25 mM Tris, 192 mM glycine) containing 10% ethanol for large proteins and 20% ethanol for small (<100 kDa) proteins at 30 V for 16 h. Membranes were blocked for 1 h and incubated in primary antibody o/n on a roller at 4 °C. Membranes were washed three times for 10 min in TBS-T (0.1% Tween in TBS), incubated in HRP-conjugated secondary antibody for 1 h at RT and chemiluminescent signal was detected on ChemiDoc MP Imaging system (Bio-Rad) operated by Bio-Rad Image Lab Touch Software (version 2.3.0.07) and analysed using Bio-Rad Image Lab Software (version 5.2.1). Antibodies used for Western Blotting are listed in Supplementary Table 1.

## RNA isolation

Cell pellets from one well on a 6-well plate were resuspended in 1 mL TRI reagent (Sigma) and incubated for 5 min at room temperature. 200 µL chloroform (Applichem) was added and the lysate was vortexed and centrifuged at max. speed (21130xg) for 15 min at 4 °C. The aqueous layer was transferred to a new tube and precipitated with 0.5 mL isopropanol. The RNA pellet was isolated by centrifugation for 30 min at 4 °C, washed with 1 mL 75% ethanol, re-centrifuged for 10 min, dried and resuspended in 70 µL RNase free water. 20 µg RNA was treated with 40 U DNaseI (Roche) for 30 min at 37 °C and subsequently purified by phenol-chloroform extraction and ethanol precipitation.

## Reverse transcription and real-time qPCR

1 µg of RNA was reverse transcribed using ProtoScript II reverse transcriptase (NEB) and oligo(dT) primer (Thermo Fisher). For reverse transcription of RNA from mESCs, NSCs and neurons, random hexamer primers (Invitrogen) were used. cDNA was diluted 1:5 in H$_2$O, 1 µL was used per 10 µL qPCR reaction. qPCR was performed on a BioRad CFX Touch cycler operated by BioRad CFX Maestro software (version 2.2) using Takyon No Rox SYBR MasterMix dTTP Blue (Eurogentec). qPCR primer sequences are indicated in Supplementary Data 5. qPCR data were analysed and plotted using GraphPad Prism (9.1.1). Experiments were performed in 3-5 biological replicates and each sample was measured in 3 technical replicates.

## Subcellular fractionation, RNA extraction and 3'end sequencing library preparation

For 3'end sequencing cells were fractionated into cytoplasm and nuclei and RNA was extracted separately. 10 million HEK293T cells were mixed with 1 million MEF cells as a spike-in control, washed once in 1xPBS and resuspended in 1 mL hypotonic buffer (10 mM HEPES pH 7.9, 1.5 mM MgCl$_2$, 10 mM KCl, 1 mM DTT, 1 mM PMSF, 0.1% Triton, 1x protease inhibitors). Cells were incubated on ice for 5 min, then dounced in a prechilled 7 mL Douncer for 10 strokes with a tight pestle to release nuclei. After centrifugation at 228xg, 4 °C for 5 min the supernatant was mixed with 200 µL 5x RIPA buffer (250 mM Tris pH 7.5, 750 mM NaCl, 5% NP-40, 2.5% deoxycholate, 1 mM PMSF, 0.1% Triton, 1x protease inhibitors) and centrifuged for 10 min at 2800xg, 4 °C. The supernatant was kept as the cytoplasmic fraction. The pellet from the first centrifugation step was resuspended in 600 µL sucrose buffer 1 (0.25 M sucrose, 10 mM MgCl$_2$, 1 mM PMSF, 1x protease inhibitors) and layered over 600 µL sucrose buffer 2 (0.35 M sucrose, 0.5 mM MgCl$_2$, 1 mM PMSF, 1x protease inhibitors). After centrifugation at 1430xg for 5 min, 4 °C, the pellet (nuclei) was resuspended in 1 mL TRI reagent and processed for RNA extraction as described above. For RNA extraction from the cytoplasmic fraction, 300 µL cytoplasmic extract were mixed with 900 µL TRI reagent LS for liquid samples (Sigma), 240 µL chloroform was added and the lysate was vortexed and centrifuged at max. speed (21130xg) for 15 min at 4 °C. The aqueous layer was transferred to a new tube and precipitated with 0.6 mL isopropanol. The RNA pellet was subsequently treated as described above. Library preparation for 3'end sequencing of polyadenylated mRNAs was performed using 3'mRNA-Seq Library Prep Kit FWD (Lexogen) according to the manufacturer's instructions. Sequencing was performed on an Illumina NextSeq 550 instrument in readmode SR75 by the Next Generation Sequencing facility at Vienna BioCenter Core Facilities (VBCF).

## RNA-seq library preparation

Cells from a 90% confluent 10 cm dish were harvested and counted, 8 million cells were mixed with 2 million Drosophila S2 cells as a spike-in control. RNA isolation was performed as described above. rRNA was depleted using NEBNext rRNA depletion kit v2 (Human/Mouse/Rat) (NEB) and libraries were prepared using NEBNext Ultra II Directional RNA Library Prep Kit for Illumina (NEB) according to the manufacturer's instructions, starting with 600 ng total RNA input. Sequencing was performed on an Illumina NovaSeq 6000 instrument in readmode SR100 by the Next Generation Sequencing facility at Vienna BioCenter Core Facilities (VBCF).

## RNA-seq data analysis

RNA-seq data from HEK293 cells from different genetic background was processed using PiGx-RNA-seq pipeline[36]. The data was mapped to the GRCh38/hg38, and dm6 versions of the human, and drosophila genomes using STAR, with the following parameters: --limitOutSJcollapsed 20000000 --limitIObufferSize=1500000000 --outFilterMultimapNmax 10 --seedPerWindowNmax 5. The mapped data was quantified using SALMON[37]. The quantified data was processed using tximport[38], and the differential expression analysis was done using DESeq2[39]. Genes with less than 5 reads in all biological replicates of one condition were filtered out before the differential analysis. The data was normalized by taking the ratio of reads mapping to the human and the Drosophila transcriptome. Variance estimation was performed separately for each condition - control sample pair. Genes were defined as differentially expressed if they had a minimum absolute log2 fold change of 1, and a BH adjusted p value less than 0.05. Browser tracks were constructed by using the size factors calculated from the spike in data.

### 3'end RNA-seq data analysis

3'end data was processed in the same way as the RNA-seq data. Instead of the Drosophila, spike-in consisted of mouse cells. The data was therefore mapped to the mm9 version of the mouse genome.

### Immunofluorescence

Glass coverslips (thickness #1.5, diameter 12 mm, sterilized by baking at 180 °C o/n) were pretreated with 10 µg/mL fibronectin (Sigma, F1141) for 3 h at room temperature or o/n at 4 °C or with 0.3 µg/mL at room temperature o/n to ensure enhanced cell adhesion. Cells were seeded onto coverslips and grown to a confluency of about 80% before fixation. Cells were washed once with 1xPBS, fixed with 4% paraformaldehyde for 10 min and washed again three times before permeabilization in 0.5% Triton X-100 for 6 min. After washing three times with PBS, cells were blocked in blocking buffer (0.1% Tween, 3% BSA in PBS) for at least 20 min at room temperature. Incubation with primary antibodies rabbit anti-DIDO1 (1:200), rabbit anti-GFP (1:1000), mouse anti-FLAG (1:700) and rat anti-Pol II pS2 (1:25) was done at 4 °C o/n, washed 3x, followed by 1:500 fluorophore-coupled secondary antibody for 1 h at room temperature in the dark. For double staining of DIDO or PHF3 with chromatin and nucleolus marks, cells were incubated with rabbit anti-H3K9ac (1:1000), anti-H3K9me3 (1:500) or anti-fibrillarin (1:200) antibody o/n at 4 °C, washed 3x, incubated with AF647-coupled anti-rabbit antibody (1:200) for 1 h at room temperature, washed 3x, incubated with mouse anti-GFP (1:200) or anti-FLAG (1:700) antibody o/n at 4 °C, washed 3x and incubated with anti-mouse secondary antibody (1:500) coupled to AF488 or AF568 respectively. All coverslips were furthermore washed two times, stained with DAPI (Sigma D8417, 1:10 000) for 5-10 min at room temperature, washed 1x with PBS and 1x with ddH$_2$O and mounted onto slides with Prolong Diamond (Invitrogen, P36961).

### Confocal and high-resolution Airyscan imaging

Immunofluorescence images were acquired with identical acquisition parameters for every experiment group (Supplementary Tables 4 and 5) using an inverse point scanning confocal Zeiss LSM980 Microscope equipped with a Zeiss Plan-Apochromat 63x/1.4 Oil DIC M27 (WD 0.19 mm) used for confocal images (Fig. 3g, Fig. 4g,h, Supplementary Figs. 2a, 15, 16d) and Plan-Apochromat 40x/1.4 Oil DIC M27 (WD 0.13 mm) objective used for all other images, running with Zeiss ZEN blue 3.3 software (version 3.3.89.0008). Sequential acquisitions of up to three channels were performed with a 405 nm (30 mW), a 488 nm laser diode (30 mW), a 561 nm DPSS laser (25 mW) and a 639 nm laser diode (25 mW) set to 0.2–15.0% excitation power together with detector gain set to 600-700 V. In Airyscan mode different secondary beam splitters were used to constrain emission wavelengths. Detection was done with a combination of GaAsP-PMT detectors for confocal images and an Airyscan 2 detector (32 GaAsP elements) for 3D colocalization analysis with z-stack spacing of 0.15-0.17 µm. DAPI staining was used to identify nuclei, laser power and detector gain were balanced for each channel to enhance signal intensity and reduce background noise. All acquisitions were done by unidirectional imaging setup with detector offset of 0, digital detector gain of 1.0 and optimized to Nyquist settings. Confocal images were acquired in 16-bit and Airyscan in 8-bit images.

### Image analysis and processing

All images were processed with Fiji/ImageJ (ImageJ 1.53c) software in parallel with the same Costes-related automatic thresholds for each channel in each individual experiment. Representative 20×20 µm field of views together with regions of interest within nuclei of 2 × 2 µm were chosen and assembled in OMERO.figure (version 4.3.2) and Adobe Illustrator (version 24.3). Airyscan images were processed for super-resolution with Airyscan filter 6. Colocalization was analysed with the Zeiss co-localization plugin of Zen 3.3 (version 3.3.89.0008) by

marking individual nuclei as regions of interest and thresholding the colocalization with Costes-related automatic threshold. The Colocalization coefficients (Manders' overlap fractions M1 & M2) for each nucleus were averaged across the whole z-stack (pixel count threshold: 3000) and each nucleus plotted individually. Nuclei from at least 2 biological replicates (total number of nuclei indicated in figure legends) were plotted in box and whiskers plots and analysed with two-tailed unpaired Student's t-test with Welch's correction or with one-way ANOVA with Brown-Forsythe & Welch's correction in Prism 9.2.0. Both analyses were performed with a confidence interval of 95% and defined statistical significance as p < 0.05. Therefore, p-values smaller than 5% were considered statistically significant and indicated with an asterisk (ns for ≥0.05; "*" for p < 0.05; "**" for p < 0.01; "***" for p < 0.001 and "****" for p < 0.0001). The Null-Hypothesis for all colocalization analysis implied that both or all three cell lines show no difference in colocalization of the two proteins of interest.

### FACS (Fluorescence activated cell sorting)

70-80% confluent cells from 6 cm dish were harvested, the pellet resuspended in 1xPBS and spun down for 5 min with 500xg at 4 °C. The pellet was washed again with PBS, resuspended in 800 µL PBS and 2.2 mL cold methanol, gently mixed by inverting the tube and incubated at −20 °C o/n. The cells were spun down at 500xg for 5 min, washed with PBS and incubated with 500 µL propidium iodide (PI) buffer (50 µg/mL PI, 10 mM Tris pH 7.5, 5 mM MgCl$_2$ freshly added 200 µg/mL RNAse A) at 37 °C for at least 30 min before FACS measurements. All samples were measured on a Bio-Rad ZE5 cell analyzer operated by Bio-Rad ZE5 Everest software (version 2.5.0.10) with excitation laser at 561 nm (50 mW, PI excitation) and a flow rate of 0.1 µL/sec. The detector voltage was adjusted for each sample to align G1 peaks and therefore PI cell cycle histograms. For each sample at least 20 000 healthy single cells were counted. Cell cycle distribution was analysed by gating G1, S and G2/M phase cells in FlowJo (version 10.8.1). FACS during cell line generation was performed on a BD FACSMelody cell sorter operated by FACSChorus software (version 1.1.20.0).

### Anti-FLAG immunoprecipitation

HEK293T cells were transfected with FLAG-DIDO constructs in 10 cm dishes at 70-80% confluency. Cells were harvested 48 h after transfection and lysed in lysis buffer (50 mM Tris-Cl pH 8, 150 mM NaCl, 0.1% Triton, 1x protease inhibitors, 1 mM PMSF, 2 mM NaF, 50 units/mL benzonase and 1 mM DTT) for 1 h at 4 °C on a rotating wheel. Lysates were centrifuged at 18,000 × g for 10 min at 4 °C. 10% of cleared lysate was kept as input, the rest was incubated with 40 µL anti-FLAG M2 magnetic beads (Sigma) for 2 h on a rotating wheel at 4 °C. Beads were washed once with lysis buffer with benzonase and DTT and four times with TBS. For SDS-PAGE and western blotting, beads were incubated with 150 ng/µL 3xFLAG peptide in TBS on the rotating wheel for 30 min to elute the proteins. Western blots were analysed using Image Lab 6.0.1 (Biorad).

### Anti-GFP immunoprecipitation

$10^7$ cells were lysed in 200 µl of lysis buffer (50 mM Tris-Cl pH 8, 150 mM NaCl, 1% Triton, 1x protease inhibitors, 1 mM PMSF, 2 mM NaF, 50 units/mL benzonase, 2 mM MgCl$_2$ and 1 mM DTT) for 30 minutes at 4 °C on a rotating wheel. Lysates were centrifuged at 18,000 × g for 10 min at 4 °C. The supernatant was transferred to a fresh tube and diluted by adding 300 µl of dilution buffer (50 mM Tris-Cl pH 8, 150 mM NaCl, 0.5 mM EDTA). 25 µl aliquot of the diluted lysate was kept as input. 25 µl slurry of agarose beads (GFP-Trap® Agarose, ChromoTek) was equilibrated by washing three times with 500 µl of dilution buffer. The diluted lysate was added to the equilibrated beads and incubated on a rotating wheel at 4 °C for 2 h to allow protein binding to the beads. After incubation, the beads were washed once with diluted lysis buffer and three times with ice-cold TBS buffer. The

proteins bound to the beads were eluted by adding 40 µl of 2x SDS sample buffer and boiling for 5 min at 95 °C. Samples were analysed by western blotting.

## Anti-Pol II pS5 immunoprecipitation

One 10 cm dish was used for each cell line. Cells were harvested and lysed in lysis buffer (as above). Protein G Dynabeads (Invitrogen) were washed twice with TBS and incubated with 5 µg of mouse anti-pS5 Pol II 4H8 for PHF3 samples or 5 µg of rat anti-pS5 Pol II 3E8 for DIDO samples for 1 h on a rotating wheel at room temperature. Beads were washed twice with TBS and cleared lysates were added for immunoprecipitation on a rotating wheel at 4 °C ON (for PHF3 samples) or at 4 °C for 2 h (for DIDO samples). After immunoprecipitation, beads were washed 6x with TBS with 1x protease inhibitors and 1 mM PMSF.

Sample preparation for mass spectrometry analysis. Beads were three times eluted with 20 µL 100 mM glycine and the combined eluates adjusted to pH 8 using 1 M Tris-Cl pH 8. Disulfide bonds were reduced with 10 mM DTT for 30 min at room temperature before adding 25 mM iodoacetamide and incubating for another 30 min at room temperature in the dark. Remaining iodoacetamide was quenched by adding 5 mM DTT and the proteins were digested with 300 ng trypsin (Trypsin Gold, Promega) overnight at 37 °C. The supernatant was transferred to a new tube, the beads were washed with another 30 µL of 2 M urea in 50 mM ABC and the wash combined with the supernatant. After diluting to 1 M urea with 50 mM ABC, 150 ng trypsin was added and incubated overnight at 37 °C in the dark. The digest was stopped by addition of 1% trifluoroacetic acid (TFA), and the peptides were desalted using C18 Stagetips[40]

## Liquid chromatography-mass spectrometry analysis

Peptides were separated on an Ultimate 3000 RSLC nano-flow chromatography system (Thermo-Fisher), using a pre-column for sample loading (Acclaim PepMap C18, 2 cm × 0.1 mm, 5 µm, Thermo-Fisher), and a C18 analytical column (Acclaim PepMap C18, 50 cm × 0.75 mm, 2 µm, Thermo-Fisher), applying a segmented linear gradient from 2% to 35% and finally 80% solvent B (80% acetonitrile, 0.1% formic acid; solvent A 0.1 % formic acid) at a flow rate of 230 nL/min over 120 min.

For pS5 IP PHF3 samples and FLAG IP with DIDO samples, eluting peptides were analysed on a Q Exactive HF-X Orbitrap mass spectrometer (Thermo Fisher), which was coupled to the column with a customized nano-spray EASY-Spray ion-source (Thermo-Fisher) using coated emitter tips (New Objective). The mass spectrometer was operated in data-dependent acquisition mode (DDA), survey scans were obtained in a mass range of 375-1500 m/z with lock mass activated, at a resolution of 120k at 200 m/z and an AGC target value of 3E6. The 8 most intense ions were selected with an isolation width of 1.6 m/z, isolation offset 0.2 m/z, fragmented in the HCD cell at 27% collision energy and the spectra recorded for max. 250 ms at a target value of 1E5 and a resolution of 30k. Peptides with a charge of +1 or >+6 were excluded from fragmentation, the peptide match feature was set to preferred, the exclude isotope feature was enabled, and selected precursors were dynamically excluded from repeated sampling for 30 seconds. Raw data were processed using the MaxQuant software package (version 1.6.0.16)[41] and the Uniprot human reference proteome (July 2018, www.uniprot.org) as well as a database of most common contaminants. The search was performed with full trypsin specificity and a maximum of three missed cleavages at a protein and peptide spectrum match false discovery rate of 1%. Carbamidomethylation of cysteine residues were set as fixed, oxidation of methionine, phosphorylation of serine, threonine and tyrosine, and N-terminal acetylation as variable modifications. For label-free quantification the "match between runs" feature and the LFQ function were activated - all other parameters were left at default. Downstream data analysis was performed using the LFQ values in Perseus (version 1.6.2.3)[41]. Mean LFQ intensities of biological replicate samples were

calculated and proteins were filtered for at least two quantified values being present in the three biological replicates. Missing values were replaced with values randomly selected from a normal distribution (with a width of 0.3 and a median downshift of 1.8 standard deviations of the sample population). To determine differentially enriched proteins we used the LIMMA package in R (version 3.5.1.) and applied the Benjamini-Hochberg correction for multiple testing to generate adjusted p-values.

For pS5 IP DIDO samples, eluting peptides were analysed on an Exploris 480 Orbitrap mass spectrometer (Thermo Fisher) coupled to the column with a FAIMS pro ion-source (Thermo-Fisher) using coated emitter tips (PepSep, MSWil) with the following settings: The mass spectrometer was operated in DDA mode with two FAIMS compensation voltages (CV) set to −45 or −60 and 1.5 s cycle time per CV. The survey scans were obtained in a mass range of 350-1500 m/z, at a resolution of 60k at 200 m/z, and a normalized AGC target at 100%. The most intense ions were selected with an isolation width of 1.2 m/z, fragmented in the HCD cell at 28% collision energy, and the spectra recorded for max. 100 ms at a normalized AGC target of 100% and a resolution of 15k. Peptides with a charge of +2 to +6 were included for fragmentation, the peptide match feature was set to preferred, the exclude isotope feature was enabled, and selected precursors were dynamically excluded from repeated sampling for 45 seconds. MS raw data split for each CV using FreeStyle 1.7 (Thermo Fisher), were analysed using the MaxQuant software package (version 2.1.0.0)[41] with the Uniprot human reference proteome (version 2022.01, www.uniprot.org), as well as a database of most common contaminants. The search was performed with full trypsin specificity and a maximum of two missed cleavages at a protein and peptide spectrum match false discovery rate of 1%. Carbamidomethylation of cysteine residues was set as fixed, oxidation of methionine, phosphorylation of serine, threonine and tyrosine, and N-terminal acetylation as variable modifications. For label-free quantification the "match between runs" only within the sample batch and the LFQ function were activated - all other parameters were left at default. MaxQuant output tables were further processed in R 4.2.0 (https://www.R-project.org) using Cassiopeia_LFQ (https://github.com/moritzmadern/Cassiopeia_LFQ). Reverse database identifications, contaminant proteins, protein groups identified only by a modified peptide, protein groups with less than two quantitative values in one experimental group, and protein groups with less than 2 razor peptides were removed for further analysis. Missing values were replaced by randomly drawing data points from a normal distribution model on the whole dataset (data mean shifted by −1.8 standard deviations, a width of the distribution of 0.3 standard deviations). To determine differentially enriched proteins we used the LIMMA package in R (version 3.5.1.)[42] and applied the Benjamini-Hochberg correction for multiple testing to generate adjusted p-values.

## Sucrose gradient ultracentrifugation

15% and 40% (w/v) sucrose solutions were prepared in 50 mM Tris-Cl pH 8, 150 mM NaCl, 1% Triton, 1 mM DTT and left to cool and degas o/n at 4 °C. Cell pellets were lysed in lysis buffer (50 mM Tris-Cl pH 8, 150 mM NaCl, 1% Triton, 1x protease inhibitors, 1 mM PMSF, 2 mM NaF, 50 units/mL benzonase and 1 mM DTT) as described under Immunoprecipitation. Protein concentrations were estimated by Bradford assay. 4 mL sucrose gradients were prepared using a Gradient mixer (Gradient Master 108; BioComp Instruments). 200 µL (5 mg/mL) protein lysate was added on top of the gradient and centrifuged at 105169xg in a SW60 swinging bucket rotor (Beckman Coulter) for 16 h at 4 °C. 100 µL fractions were collected, mixed with SDS sample buffer and analysed by Western blotting.

## Protein purification

Full-length PHF3 was expressed and purified from insect cells as previously described[3]. Pol II was purified from pig thymus as previously

described[43]. Pig thymus was sourced from animals approved for food consumption through an officially approved facility in Sieghartskirchen, Lower Austria. Pol II was labeled with Alexa Fluor 488, Conjugation Kit (Fast) · Lightning-Link (abcam) according to the manufacturer's instructions and purified over a Superdex 200 Increase 3.2/200 column (Cytiva). mCherry-tagged PHF3 constructs and mCherry- or mEGFP-tagged DIDO constructs were expressed *in E. coli* Rosetta2 (DE3) cells and purified by affinity chromatography using HisTrap HP column (Cytiva) equilibrated in 25 mM Tris-Cl pH 7.4, 500 mM NaCl, 20 mM imidazole, followed by size exclusion chromatography using Superdex 75 (Cytiva) equilibrated in 25 mM Tris-Cl pH 7.4, 100 mM NaCl, 10% glycerol and 1 mM DTT.

## In vitro condensate formation

4-well glass bottom slides (Ibidi) were coated with 1% PF127 (Sigma Aldrich) overnight at 4 °C and washed twice with 25 mM Tris-Cl pH 7.5, 50 mM NaCl, 1 mM DTT, and 10% v/v PEG6000. Pol II was prepared by mixing Alexa labeled and unlabeled protein at 1:5 ratio. Protein samples were loaded onto glass slides, mixed with the buffer to reach the final concentration of 20 mM Hepes pH 7.4, 150 mM NaCl, 1 mM TCEP, and imaged within 15 to 45 min. Imaging was performed on Zeiss Axio Observer Z1 with a 60× oil immersion objective using Zen Blue software (version 3.3.89.0008). Condensate size was analysed using Fiji (ImageJ 1.53c).

## Differentiation of mESCs into NSC, neurons and astrocytes

mESCs were maintained in standard DMEM-FBS/Lif medium on gelatin. Prior to differentiation, they were grown in N2B27 + 2i/Lif medium for 2 passages. Differentiation into neural stem cells (NSCs) and later into neurons was adapted from a previously described protocol[44]. Briefly, 10000 mESCs/cm² were seeded on gelatin-coated 10 cm dishes and cultured for 7 days in N2B27 medium. After 7 days, $2-5 \times 10^6$ cells were transferred to non-gelatinized T75 flasks in NS-N2B27 medium (N2B27 medium supplemented with 10 ng/mL EGF and 10 ng/mL FGF2) and grown for 2–4 days to form aggregates in suspension. The cell aggregates were then collected by centrifugation ($105 \times g$ for 30 s) and transferred to fresh gelatin-coated T75 flasks and grown in NS-N2B27 medium. After 3 to 7 days, cells displayed NSCs morphology. For neuronal differentiation, NSCs were seeded in NS-N2B27 medium at a density of 25000 cells/cm² on laminin-coated glass coverslips in 24-well plates for immunofluorescence and 6-well plates for RNA isolation. The day after, the medium was replaced with N2B27 medium supplemented with only 5 ng/mL FGF2. Cells grown on glass coverslips were then fixed for immunofluorescence, while cells grown in 6-well plates were harvested for RNA isolation at the indicated time points. Since cell quantification was not possible due to the organization of WT differentiated cells into tight aggregates, to quantify the differences between Phf3 and Dido WT and KO cells upon neuronal differentiation we manually counted by fluorescence microscopy all TuJ1 positive cell aggregates (referred to as "neuronal clumps") on the glass coverslips. For differentiation of NSCs into astrocytes, NSCs were seeded in N2B27 medium supplemented with 1% FBS at a density of 50000 cells/cm² on gelatin coated glass coverslips in 24-well plates. Cells were fixed after 5 days and samples were processed for immunofluorescence. For immunofluorescence, both neurons and astrocytes were washed with PEM buffer (100 mM Pipes, 5 mM EGTA, 2 mM $MgCl_2$, pH 6.8) prior to fixation in 4% PFA.

## Reporting summary

Further information on research design is available in the Nature Portfolio Reporting Summary linked to this article.

## Data availability

The source data are provided in this paper. 3′end mRNA-sequencing data generated in this study have been deposited in ArrayExpress under accession code: E-MTAB-12757. RNA-sequencing data generated in this study have been deposited in ArrayExpress under accession code: E-MTAB-12782. The processed RNA-seq data are provided in Supplementary Data 4. Mass spectrometry data have been deposited to the ProteomeXchange Consortium via the PRIDE partner repository[45] with the dataset identifier PXD039540 for FLAG IP, PXD039537 for Pol II pS5 IP in PHF3 mutant cells and PXD039567 for Pol II pS5 IP in DIDO mutant cells. The processed mass spectrometry data are provided in Supplementary Data 1-3. Oligonucleotides used in the study are provided in Supplementary Data 5. Genomic DNA sequences were retrieved from Ensembl [https://www.ensembl.org][46]. Protein sequences were retrieved from UniProt [https://www.uniprot.org]. Source data are provided in this paper.

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

## Acknowledgements

We thank Martin Leeb for sharing the Dido1 KO mESC line; the VBCF Next Generation Sequencing facility; the Max Perutz Labs Mass Spectrometry Facility, Markus Hartl, Dorothea Anrather, WeiQiang Chen, and Natascha Hartl for mass spectrometry sample processing, data acquisition and analysis; the Max Perutz Labs Biooptics facility. This work was funded by the Austrian Science Fund (P31546 to D.S.)

## Author contributions

J.B. generated cell lines, performed and analysed RT-qPCR, performed RNA-seq experiments, performed and analysed co-immunoprecipitation assays, sucrose gradient ultracentrifugation, proliferation assays, and analysed mass spectrometry data. V.F. designed and performed the analysis of NGS data. L.A. generated endogenously tagged PHF3 ΔIDR cell line and performed RNA-seq experiments. L.W. generated PHF3-GFP DIDO full KO and DIDO-mEGFP PHF3-mScarlet cell lines and performed microscopy and FACS experiments. M.B. performed neuronal differentiation experiments. R.S. performed microscopy experiments. J.G. performed RT-qPCR. H.O. and E.F. purified DIDO and PHF3 LLPS constructs and performed in vitro condensate formation assays. X.S. performed 3' end sequencing and supervised protein purification. A.P. and B.Z. analysed LLPS properties. A.A. designed and supervised sequencing data analysis. D.S. conceived the study, performed, supervised, analysed experiments, and wrote the manuscript.

## Competing interests

The authors declare no competing interests.
