## [Peer Review File · Nature Communications]

The SPOC proteins DIDO3 and PHF3 co-regulate gene expression and neuronal differentiationREVIEWER COMMENTS

Reviewer #1 (Remarks to the Author):

Benedum et al Nature communication 2023

This is an excellent multi-technique driven study into transcription regulation by two paralogue proteins which regulate neuronal genes. The fusion of genomics/mass spec/imaging provides robust data supporting the conclusions of the authors. Through this work they are able to determine a multi-layered approach to regulating gene expression, and link the regulation to direct interactions with RNA Polymerase II through phase separation.

The experiments are performed to a high standard throughout, the manuscript is well-written and easy to follow, and data are largely presented in a clear manner.

I believe this work is of high interest to both the neuronal differentiation field but also the wider transcription community.

I have the following comments which should be addressed:

(1) For the interactome analysis: Over expression of proteins can drive interactions which may be minor or non-specific. It would have been good to compare to endogenous protein interactions for one of the proteins.

(2) Were the expression levels of the FLAG constructs equal? Once again this could impact the interactome analysis. I could only find co-IP gels whereas it would be better to show the expression levels on a gel.

(3) I believe it would be interesting to visually display similarities and differences between the interactomes – for example, in a Venn diagram? This will make it clear on the numbers of specific and shared interactions.

(4) The following is stated: Endogenously tagged DIDO3-mEGFP and PHF3-mScarlet in the previously generated double-tagged cell line (Supplementary Fig. 9) showed uneven distribution throughout chromatin, with highly compacted areas showing slightly weaker DIDO3 signal, suggesting that they may preferentially localize to euchromatin (Fig. 3g). Can you enhance this conclusion by staining with euchromatin marks?

(5) It is stated (Abstract and discussion) that: Here we show that DIDO3 and PHF3 form a complex that bridges the Pol II elongation machinery with chromatin and RNA processing factors, and tethers Pol II in a phase-separated microenvironment. I am interested in the concept of tethering and phase-separation. I think more explanation is required as to how those two elements go together? It is often assumed that LLPS will keep components together therefore why tether?

(6) Figure text size (especially figure 1) – The text size is very small and difficult to read. I would enhance the text size. The same applies for some of the volcano plots. There are many small labels so the effectiveness of the figure is lost.

(7) Fig 3g – The staining morphology is interesting because the regions lacking DIDO3 staining do not appear to be the nucleolus (based on DAPI stain) – can you comment on this?

(8) Figure 4b – please scale both plots to 1.

(9) Sup Fig 2 – there appears to be a strong nuclear edge staining effect – can you comment on this? Is the protein tethered to the lamina/membrane?

Reviewer #2 (Remarks to the Author):

SPOC (Spen paralog and ortholog C-terminal) is a 15- to 20-kDa protein domain that recognizes phosphorylated serines. The SPOC in human DIDO3 and PHF3 can bind to phosphoserines at the C-terminal domain (CTD) of the largest subunit of RNA Polymerase II (Pol II), the complex that catalyzes the synthesis of RNA at genes and enhancers. In human, the Pol II CTD consist of 52 YSPTSPS repeats that change phosphorylation status, and recruit RNA processing complexes and elongation and termination factors as transcription proceeds from initiation to termination. Besides the SPOC domain, DIDO3 and PHF3 contain Plant Homeodomain (PHD) motif that binds to histones, and TFIIS like domain (TLD) that can compete with TFIIS on Pol II binding. These domains allow DIDO and PHF3 to form multi-complex connection at the chromatin. Recently, the Slade lab published two studies in Nature Communications (Appel et al., 2021, 2023) showing that DIDO and PHF3 associate with ser-2-phosphorylated Pol II CTD, and that DIDO and PHF3 influence mRNA levels of neuronal genes.

In this study, Benedum and co-workers (Slade lab) investigate the collaboration of DIDO3 and PHF3 and their effect on transcription. Knock-out of PHF3 or its SPOC is shown to induce isoform switching of DIDO (isoform 1 to isoform 3), and DIDO and PHF3 are shown to co-localize in phase-separated Pol II condensates in vitro. The study further characterizes interacting partners of DIDO with mass spec and identifies changes in mRNA levels caused by DIDO mutations. The main message of this study is somewhat vague and there is a lack of mechanistic insights into how DIDO and PHF3 affect the transcription machinery. My major concerns relate to the claimed novelty of the DIDO3-PHF3 co-regulation of transcription, and the lack of mechanistic insights into transcriptional coordination and neuronal differentiation. In essence, many of the results presented here are similar (in some panels identical) to what the group published in the earlier studies. The novelty and mechanisms need to be clarified and strengthened before I can recommend the study for publication. Please see my major concerns and suggestions below.

Major concerns:

- 1) The conceptual novelty of this study is either vague or unclear. The title of the study is: "The SPOC Proteins DIDO3 and PHF3 co-regulate gene expression in neuronal differentiation". The earlier study by this group (<https://www.nature.com/articles/s41467-023-35853-1>) already showed DIDO3 and PHF3 to regulate Pol II (see e.g. Figure 9 which depicts DIDO3 and PHF3 on chromatin contacting Pol II and is close to identical with the Figure 2g in this study). Figure 5 in this study has several panels that are identical to panels in Figure 6 of the previous study. What exactly is the novelty here?
- 2) The study makes the claim that DIDO3 and PHF3 control transcription in neuronal differentiation. Both DIDO3 and PHF3 seem to have a global effect on transcription (see Figure 4f for PHF3, Figure 5c uppermost panel for DIDO3). If the whole transcription is downregulated, how likely is a specific regulation of neuronal development? The mechanistic link from DIDO or PHF3 to transcription and neuronal differentiation remains to be shown (see related points 3 and 5).
- 3) The claimed transcriptional regulation is investigated via mRNA levels. In the previous study, the group also generated TT-seq and PRO-seq data. Could these existing and/or new data with direct measures on nascent (PRO-seq) and newly synthesized (TT-seq) datasets be used to dissect how PHF3 and DIDO affect Pol II progression through genes? mRNA levels do not allow detecting changes in Pol II progression and regulation.
- 4) In the cell growth assay, DIDO mutants show slower growth rate, particularly at days 3-4. These mutants contain increased proportion of cells in G2/M and reduced proportion in S phase. Can the ~10% increase of cells in G2/M explain the reduced growth rates or is cell death involved? Understanding how DIDO affects cell proliferation (and apoptosis) is crucial to be able to comprehend its role in neuronal differentiation.
- 5) Overall, the link of DIDO and PHF3 to neuronal differentiation needs strengthening:
 - a. Lack of DIDO3 and PHF3 seem to cause clear defects in neuronal differentiation in the in vitro

assay of mESCs -> NCSs -> neurons (Figure 8c). However, also DAPI (cell count) is clearly reduced. Given the effect of DIDO3 to cell proliferation, and the apparent lower cell count at the slide, the effect could be caused by a general block in proliferation (not specific to neurons).

b. Multiple mechanisms could cause the 'reduced neuronal bundles'. Is the effect of DIDO3 and/or PHF3 to neuronal development carried via Pol II regulation, i.e. transcription as claimed in the title of the study? Is there a specific time in the development when the defects in neuronal development start to appear?

c. Could the authors induce neuronal differentiation model to address in which stages of neuronal differentiation DIDO3 and/or PHF3 cause their defects?

6) mRNA-seq is stated to show "downregulation of...17412 genes in PHF3 deltaIDR" (Figure 4f). Is this number reasonable given that humans have 20 000 genes, of which 10 000 – 12 000 are generally expressed in a given cell type? Do you perhaps list transcripts (instead of genes) or is e.g. one condition shallowly sequenced giving zeros that cannot be normalized? Something seems off.

Minor comments:

1) The title of the legend of figure 5 is somewhat hard to read and seems conceptually incorrect: "RNA-seq analyses of single and combined ... cell lines".

2) The title of the legend of Figure 3 "...form condensates that regulate localization". Localization of what? Do you mean that the condensates are in different locations? In general, it is hard to couple the condensate formation to a particular function.

3) The clearest part of this study is the characterization of DIDO3 interacting proteins and showing which motifs are required for each interaction (Figure 2). To me, this gets closest to characterizing how DIDO can couple Pol II to chromatin and transcriptional regulation. In Figure 2d (lower left) Pol II pSer2, Pol II pSer5 and Pol II pSer7 signal is stronger in the presence of DIDO3 and DIDO3 dN than in any other condition. Does the presence of DIDO SPOC stabilize Pol II ser phosphorylation?

We would like to thank the reviewers for taking their time to provide constructive criticism of our manuscript, which helped us make necessary improvements as specified below. In blue we have provided point-by-point answers to reviewers' comments and in yellow we have highlighted all changes in the manuscript text. Prompted by Reviewer 2, we revisited the RNA-seq analyses and found variable efficiency of rRNA depletion of the spike-in transcriptome (causing a higher amount of RNA mapping to the repetitive elements in the Drosophila genome, used for spike-in normalization), which made us redo the whole analysis by normalizing the data to the ratio of uniquely mapping reads (human genome vs Drosophila). The samples that were most affected by the batch effect are PHF3 Δ IDR, DIDO Long Isoform KO, DIDO Δ SPOC and PHF3 Δ SPOC DIDO Δ SPOC. Figures that were changed as a result are: Figure 4e,f; Figure 5; Figure 7a; Supplementary Figures 20-22.

REVIEWER COMMENTS

Reviewer #1 (Remarks to the Author):

Benedum et al Nature communication 2023

This is an excellent multi-technique driven study into transcription regulation by two paralogue proteins which regulate neuronal genes. The fusion of genomics/mass spec/imaging provides robust data supporting the conclusions of the authors. Through this work they are able to determine a multi-layered approach to regulating gene expression, and link the regulation to direct interactions with RNA Polymerase II through phase separation.

The experiments are performed to a high standard throughout, the manuscript is well-written and easy to follow, and data are largely presented in a clear manner.

I believe this work is of high interest to both the neuronal differentiation field but also the wider transcription community.

I have the following comments which should be addressed:

(1) For the interactome analysis: Over expression of proteins can drive interactions which may be minor or non-specific. It would have been good to compare to endogenous protein interactions for one of the proteins.

To address this, we used endogenously GFP-tagged DIDO3 and DIDO3 Δ SPOC cell lines and performed anti-GFP immunoprecipitation to verify the interacting partners. These results are now included in Supplementary Fig. 2c.

(2) Were the expression levels of the FLAG constructs equal? Once again this could impact the interactome analysis. I could only find co-IP gels whereas it would be better to show the expression levels on a gel.

Figure 2d on the left side shows input samples (whole cell lysates of FLAG-tagged constructs after transient transfection in HEK293 cells). DIDO3 Δ N, DIDO2, DIDO2 Δ N and DIDO1 Δ N show lower expression levels compared to other constructs. However, IP samples on the right side show that the amount of bait bound to the beads is very similar for all constructs except for DIDO2 and DIDO2 Δ N,

which may have affected the binding of PARP2 and PAF1 that are expected to interact with DIDO2 as they also show interaction with the shorter isoform DIDO1. We have included this statement in the manuscript. It is not possible to specifically IP the endogenous DIDO2 isoform, which is why we can only show results from transient overexpression.

(3) I believe it would be interesting to visually display similarities and differences between the interactomes – for example, in a Venn diagram? This will make it clear on the numbers of specific and shared interactions.

Venn diagrams are now included in Supplementary Fig. 5.

(4) The following is stated: Endogenously tagged DIDO3-mEGFP and PHF3-mScarlet in the previously generated double-tagged cell line (Supplementary Fig. 9) showed uneven distribution throughout chromatin, with highly compacted areas showing slightly weaker DIDO3 signal, suggesting that they may preferentially localize to euchromatin (Fig. 3g). Can you enhance this conclusion by staining with euchromatin marks?

We performed immunofluorescence analysis using H3K9ac as a euchromatin histone mark (Supplementary Fig. 15), based on which we conclude that both DIDO3 and PHF3 are found in euchromatic regions.

(5) It is stated (Abstract and discussion) that: Here we show that DIDO3 and PHF3 form a complex that bridges the Pol II elongation machinery with chromatin and RNA processing factors, and tethers Pol II in a phase-separated microenvironment. I am interested in the concept of tethering and phase-separation. I think more explanation is required as to how those two elements go together? It is often assumed that LLPS will keep components together therefore why tether?

In the Abstract, tethering refers to LLPS-driven assembly, as we showed that DIDO3 and PHF3 sequester Pol II into condensates. In the Discussion, for RNA processing factors we cannot make statements regarding the mechanism of tethering as we have not analysed their PHF3-dependent condensation *in vitro* or clustering in cells.

(6) Figure text size (especially figure 1) – The text size is very small and difficult to read. I would enhance the text size. The same applies for some of the volcano plots. There are many small labels so the effectiveness of the figure is lost.

We now increased the font size.

(7) Fig 3g – The staining morphology is interesting because the regions lacking DIDO3 staining do not appear to be the nucleolus (based on DAPI stain) – can you comment on this?

As the reviewer points out, DIDO3 is not only excluded from nucleoli but also from some other nuclear regions. We performed immunofluorescence microscopy using H3K9me3 (heterochromatin marker) and fibrillarin (nucleolar marker) to better characterize DIDO3 localization. We found that DIDO3 and PHF3 are also excluded from other heterochromatic regions, in accordance with H3K9ac staining (see reply to comment 4). Interestingly, H3K9me3 showed altered distribution in PHF3 Δ IDR cells with clearly defined clusters resembling mouse chromocenters, suggesting that loss of PHF3 IDR results in

genome reorganization (Fig. 4g,h). Enhanced H3K9me3 clustering in PHF3 Δ IDR cells is concordant with gene downregulation observed by RNA-seq (Fig. 4f).

(8) Figure 4b – please scale both plots to 1.

This was modified.

(9) Sup Fig 2 – there appears to be a strong nuclear edge staining effect – can you comment on this? Is the protein tethered to the lamina/membrane?

The protein is not tethered to the lamina as we did not find any lamina interactors in the mass spectrometry analysis. This seems to be an overexpression artefact.

Reviewer #2 (Remarks to the Author):

SPOC (Spen paralog and ortholog C-terminal) is a 15- to 20-kDa protein domain that recognizes phosphorylated serines. The SPOC in human DIDO3 and PHF3 can bind to phosphoserines at the C-terminal domain (CTD) of the largest subunit of RNA Polymerase II (Pol II), the complex that catalyzes the synthesis of RNA at genes and enhancers. In human, the Pol II CTD consist of 52 YSPTSPS repeats that change phosphorylation status, and recruit RNA processing complexes and elongation and termination factors as transcription proceeds from initiation to termination. Besides the SPOC domain, DIDO3 and PFH3 contain Plant Homeodomain (PHD) motif that binds to histones, and TFIIIS like domain (TLD) that can compete with TFIIIS on Pol II binding. These domains allow DIDO and PHF3 to form multi-complex connection at the chromatin. Recently, the Slade lab published two studies in Nature Communications (Appel et al., 2021, 2023) showing that DIDO and PHF3 associate with ser-2-phosphorylated Pol II CTD, and that DIDO and PHF3 influence mRNA levels of neuronal genes.

In this study, Benedum and co-workers (Slade lab) investigate the collaboration of DIDO3 and PHF3 and their effect on transcription. Knock-out of PHF3 or its SPOC is shown to induce isoform switching of DIDO (isoform 1 to isoform 3), and DIDO and PHF3 are shown to co-localize in phase-separated Pol II condensates in vitro. The study further characterizes interacting partners of DIDO with mass spec and identifies changes in mRNA levels caused by DIDO mutations. The main message of this study is somewhat vague and there is a lack of mechanistic insights into how DIDO and PHF3 affect the transcription machinery. My major concerns relate to the claimed novelty of the DIDO3-PHF3 co-regulation of transcription, and the lack of mechanistic insights into transcriptional coordination and neuronal differentiation. In essence, many of the results presented here are similar (in some panels identical) to what the group published in the earlier studies. The novelty and mechanisms need to be clarified and strengthened before I can recommend the study for publication. Please see my major concerns and suggestions below.

Major concerns:

1) The conceptual novelty of this study is either vague or unclear. The title of the study is: “The SPOC Proteins DIDO3 and PHF3 co-regulate gene expression in neuronal differentiation”. The earlier study

by this group (<https://www.nature.com/articles/s41467-023-35853-1>) already showed DIDO3 and PHF3 to regulate Pol II (see e.g. Figure 9 which depicts DIDO3 and PHF3 on chromatin contacting Pol II and is close to identical with the Figure 2g in this study). Figure 5 in this study has several panels that are identical to panels in Figure 6 of the previous study. What exactly is the novelty here?

There are several novel findings presented in this study, including:

1) Despite having a similar domain architecture and a common Pol II CTD recruitment platform, we showed that SPOC-containing paralogues DIDO and PHF3 have compensatory, co-regulatory and specific functions in transcription.

2) Mechanistically, we show that compensatory functions can be ascribed to isoform switching, whereby deletion of PHF3 results in DIDO isoform switching from the smallest DIDO1 isoform to the largest DIDO3 isoform, which contains the SPOC domain and can thus compensate for PHF3 loss. This is the first example of paralogue buffering through isoform switching rather than a change in overall gene expression levels.

3) PHF3 and DIDO have co-regulatory functions on the genes required for neurogenesis and neuronal differentiation and are both essential for neuronal differentiation of stem cells. Previously this was shown only for PHF3 (Appel et al, 2021).

Given that we studied transcriptional co-regulation by PHF3 and DIDO using double mutant cell lines, it was important to include previously published data with single mutant cell lines (Appel et al, 2021, 2023, Nature Comms) for the sake of easier comparison (otherwise the readers would have to look at two different articles at the same time).

PHF3 interactome was previously published (Appel et al, 2021) and in this manuscript we analysed DIDO interactomes. As this manuscript focuses on dissecting common and differential functions of these two paralogues, we also included a schematic depicting that they both contact Pol II but also have distinct binding partners. Given that a similar schematic was already used in Appel et al, 2023, as the reviewer points out, we have now included a new schematic.

2) The study makes the claim that DIDO3 and PHF3 control transcription in neuronal differentiation. Both DIDO3 and PHF3 seem to have a global effect on transcription (see Figure 4f for PHF3, Figure 5c uppermost panel for DIDO3). If the whole transcription is downregulated, how likely is a specific regulation of neuronal development? The mechanistic link from DIDO or PHF3 to transcription and neuronal differentiation remains to be shown (see related points 3 and 5).

Our RNA-seq data shows that PHF3 and DIDO do not have a global effect on transcription but regulate a subset of genes, with 444 genes upregulated in PHF3 KO (Fig. 5b) and 2264 and 984 genes downregulated in DIDO Long Isoform KO and full KO respectively (Supplementary Fig. 20a, Fig. 5a). GO analysis showed that neuronal genes are enriched among deregulated genes in single KOs (Appel et al, 2023 and Fig. 7a and Supplementary Fig. 21 in this manuscript). Moreover, we previously showed that Phf3 KO mESCs can differentiate into astrocytes but not neurons, suggesting that Phf3 specifically regulates differentiation into the neuronal lineage (Fig. 8f, Appel et al, 2021). We found the same to be true for Dido KO mESCs and we included this data in the revised manuscript (Fig. 8c). Finally, we found that depletion of both Phf3 and Dido in mESCs leads to deregulated expression of transcription factors that are specifically required for neuronal fate specification, such as *Ascl1* and *Pou3f2* (Fig. 8e). All combined, our results support a specific function of Phf3 and Dido in regulating neuronal gene expression and neuronal differentiation. In the case of PHF3 Δ DR mutant (Fig. 4f), which shows

downregulation in gene expression, we found enrichment of neuronal genes. New analysis of H3K9me3 by immunofluorescence microscopy reveals a striking change in H3K9me3 distribution in PHF3 Δ IDR (Fig. 4h), from rather disperse to more defined clusters, suggesting that loss of PHF3 IDR causes a change in genome organization and increased heterochromatin formation, in accordance with the observed gene downregulation based on RNA-seq.

3) The claimed transcriptional regulation is investigated via mRNA levels. In the previous study, the group also generated TT-seq and PRO-seq data. Could these existing and/or new data with direct measures on nascent (PRO-seq) and newly synthesized (TT-seq) datasets be used to dissect how PHF3 and DIDO affect Pol II progression through genes? mRNA levels do not allow detecting changes in Pol II progression and regulation.

In Appel et al, 2021 and 2023 we analysed the effect on PHF3 and DIDO KO on Pol II progression through genes and found that both proteins regulate Pol II pause release, with PHF3 acting as a positive regulator and DIDO as a negative regulator of pause release. Moreover, we found that PHF3 positively regulates elongation rate.

4) In the cell growth assay, DIDO mutants show slower growth rate, particularly at days 3-4. These mutants contain increased proportion of cells in G2/M and reduced proportion in S phase. Can the ~10% increase of cells in G2/M explain the reduced growth rates or is cell death involved? Understanding how DIDO affects cell proliferation (and apoptosis) is crucial to be able to comprehend its role in neuronal differentiation.

We did not observe apoptosis in DIDO KO cells based on cleaved PARP1 used as an apoptotic marker (Supplementary Fig. 16e). This suggests that apoptosis does not contribute to reduced proliferation in DIDO KO. In addition, we performed RT-qPCR analysis to validate RNA-seq data showing that DIDO positively regulates the expression of E2F1, which is required for proliferation (Fig. 6e).

5) Overall, the link of DIDO and PHF3 to neuronal differentiation needs strengthening:
a. Lack of DIDO3 and PHF3 seem to cause clear defects in neuronal differentiation in the in vitro assay of mESCs -> NCSs -> neurons (Figure 8c). However, also DAPI (cell count) is clearly reduced. Given the effect of DIDO3 to cell proliferation, and the apparent lower cell count at the slide, the effect could be caused by a general block in proliferation (not specific to neurons).
b. Multiple mechanisms could cause the 'reduced neuronal bundles'. Is the effect of DIDO3 and/or PHF3 to neuronal development carried via Pol II regulation, i.e. transcription as claimed in the title of the study? Is there a specific time in the development when the defects in neuronal development start to appear?

c. Could the authors induce neuronal differentiation model to address in which stages of neuronal differentiation DIDO3 and/or PHF3 cause their defects?

a. We performed cell cycle profiling by propidium iodide staining and FACS analysis for WT, Phf3 KO and Dido1 heterozygous KO mESCs, grown in regular DMEM+FBS medium used to maintain the cells or in N2B27 (+Lif+2i) medium, which was used for two passages prior to the neural differentiation experiment. We could not detect any major differences between the different cell lines at the ES stage, suggesting that Dido heterozygous depletion in mESCs does not impair replication and proliferation. These experiments are included in Supplementary Fig. 24.

We observed major cell death only at the last stage of differentiation, from NSCs to neurons, for both Phf3 KO and Dido KO cells, which is reflected in the DAPI staining. This is not a proliferation defect because terminally differentiated cells do not divide anymore. We seeded the same number of NSCs for WT, Phf3 KO and Dido heterozygous KO for terminal neuronal differentiation (as well as astrocyte differentiation). WT cells that successfully differentiated into neurons migrated and organized themselves into neuronal bundles, but they did not proliferate. Phf3 KO and Dido heterozygous KO cells did not differentiate properly and failed to connect/migrate into bundles and most likely died because of this. Neuronal cell death was previously reported to be caused by axonal failure and lack of synapsis (PMID: 20480262).

b/c. We have provided RT-qPCR data in Fig. 8e showing that Dido heterozygous depletion results in sustained upregulation of key neuronal transcription factors in stem cells, neural stem cells and neurons. These neuronal transcription factors must be tightly regulated for the differentiation of progenitors along the neuronal lineage and their premature expression was previously shown to interfere with terminal neuronal differentiation (PMID: 25520623, 24243019, 15995704). We also found that Dido expression levels are increased in neural stem cells and neurons compared to stem cells, in accordance with its function in regulating neural fate, which we previously also reported for Phf3 (included as Fig. 8f). Loss of Phf3 leads to consistent upregulation of Dido at all three cell stages and vice versa (Fig. 8f). Overall, our data suggest that Dido and Phf3 promote neuronal fate specification by regulating the timing of the expression of neuronal transcription factor as stem cells are committed to neural fate.

6) mRNA-seq is stated to show “downregulation of...17412 genes in PHF3 deltaIDR” (Figure 4f). Is this number reasonable given that humans have 20 000 genes, of which 10 000 – 12 000 are generally expressed in a given cell type? Do you perhaps list transcripts (instead of genes) or is e.g. one condition shallowly sequenced giving zeros that cannot be normalized? Something seems off.

Thank you for this remark, which prompted us to revisit the RNA-seq analyses and find variable efficiency of rRNA depletion of the spike-in transcriptome (causing a higher amount of RNA mapping to the repetitive elements in the Drosophila genome, used for spike-in normalization), which made us redo the whole analysis by normalizing the data to the ratio of uniquely mapping reads (human genome vs Drosophila). The sample that was most affected by the batch effect is PHF3 Δ IDR, which now shows downregulation of 3891 genes.

Minor comments:

1) The title of the legend of figure 5 is somewhat hard to read and seems conceptually incorrect: “RNA-seq analyses of single and combined ... cell lines”.

We changed the title into: RNA-seq analysis of single mutants and combined mutants of DIDO and PHF3 in HEK293T cell lines.

2) The title of the legend of Figure 3 “...form condensates that regulate localization”. Localization of what? Do you mean that the condensates are in different locations? In general, it is hard to couple the condensate formation to a particular function.

We changed the title by removing localization.

3) The clearest part of this study is the characterization of DIDO3 interacting proteins and showing which motifs are required for each interaction (Figure 2). To me, this gets closest to characterizing how DIDO can couple Pol II to chromatin and transcriptional regulation. In Figure 2d (lower left) Pol II pSer2, Pol II pSer5 and Pol II pSer7 signal is stronger in the presence of DIDO3 and DIDO3 dN than in any other condition. Does the presence of DIDO SPOC stabilize Pol II ser phosphorylation?

Thank you for this remark. Indeed, DIDO stabilizes Pol II CTD phosphorylation as we observe increased phosphorylation upon DIDO overexpression and reduced phosphorylation in DIDO KO cells. These phenotypes are currently studied in more detail as part of another project.

REVIEWERS' COMMENTS

Reviewer #1 (Remarks to the Author):

I am happy with the changes made by the authors and I thank them for the effort to address the comments.

Reviewer #2 (Remarks to the Author):

The revised manuscript now supports the conclusions made:

- 1) The re-analyzed RNA-seq data suggests that disruption of neither PHF3 nor DIDO causes genome-wide changes to transcription, giving credibility to their specific role in coordinating transcription to promote neuronal differentiation.
- 2) Visualization of H3K9me3 distribution via microscopy, and detailed analyses of cell proliferation, support PHF3 and DIDO -driven chromatin reorganization and differentiation.
- 3) While terminally differentiated WT cells re-organised into bundles, PHF3 KO or DIDO HZ cells failed to connect, migrate and form clusters.
- 4) Updated figures and wording provide the needed clarity for the study.

With the new data and analyses, I'm happy to support the publication of this study. This work describes DIDO protein-interactomes with mass spec, tracks the influence of DIDO and PHF3 to RNA-expression, and uncovers their interplay during neuronal differentiation, likely at the stage of fate-determination or neuronal organization.